# Carbohydrate-aromatic interface and molecular architecture of lignocellulose

Alex Kirui [1,5], Wancheng Zhao[1,5], Fabien Deligey[1,5], Hui Yang [2], Xue Kang[1,4], Frederic Mentink-Vigier [3] & Tuo Wang [1✉]

Plant cell walls constitute the majority of lignocellulosic biomass and serve as a renewable resource of biomaterials and biofuel. Extensive interactions between polysaccharides and the aromatic polymer lignin make lignocellulose recalcitrant to enzymatic hydrolysis, but this polymer network remains poorly understood. Here we interrogate the nanoscale assembly of lignocellulosic components in plant stems using solid-state nuclear magnetic resonance and dynamic nuclear polarization approaches. We show that the extent of glycan-aromatic association increases sequentially across grasses, hardwoods, and softwoods. Lignin principally packs with the xylan in a non-flat conformation via non-covalent interactions and partially binds the junction of flat-ribbon xylan and cellulose surface as a secondary site. All molecules are homogeneously mixed in softwoods; this unique feature enables water retention even around the hydrophobic aromatics. These findings unveil the principles of polymer interactions underlying the heterogeneous architecture of lignocellulose, which may guide the rational design of more digestible plants and more efficient biomass-conversion pathways.

[1] Department of Chemistry, Louisiana State University, Baton Rouge, LA 70803, USA. [2] Department of Biology, Pennsylvania State University, University Park, PA 16802, USA. [3] National High Magnetic Field Laboratory, Tallahassee, FL 32310, USA. [4]Present address: Institute of Drug Discovery Technology, Ningbo University, 315211 Ningbo, Zhejiang, China. [5]These authors contributed equally: Alex Kirui, Wancheng Zhao, Fabien Deligey. ✉email: tuowang@lsu.edu

With solar energy and carbon dioxide transformed into carbohydrate-rich cell walls, terrestrial plants constitute 80% of the biomass distributed in the biosphere[1]. The secondary cell wall is a lignocellulosic composite deposited once the cellular expansion has ceased, which has evolved into a major source of biopolymers and biofuels[2,3]. Lignification mechanically strengthens secondary walls; however, the presence of these intractable polyphenols and their association with carbohydrate components contributes to the biomass recalcitrance that renders the feedstock resistant to enzymatic hydrolysis during its conversion to liquid transportation fuel[4,5]. To cost-effectively access structural polysaccharides for ethanol fermentation, vast efforts have been dedicated to tailoring plants to produce more digestible walls and optimizing deconstruction procedures in biorefineries[6–8]. These efforts have not yet reached the full potential due to our limited understanding of cell wall architecture.

The secondary cell wall is assembled by carbohydrate and aromatic constituents, with remarkable complexity and variability. Each elementary cellulose microfibril contains eighteen 1,4-β-glucan chains, which are held together by a hydrogen-bonding network[9,10]. The exact organization of these glucan chains is unresolved, but recent density functional theory (DFT) calculations suggest a six-layered organization, likely with 2, 3, 4, 4, 3, and 2 chains in each layer (Fig. 1a)[11]. Elementary microfibrils frequently coalesce, forming large fibrils that often span across tens of nanometers[12,13]. Hemicelluloses, such as xylan, glucuronoxylan, arabinoxylan, and glucomannan, are highly variable in their monosaccharide composition and linkage pattern. Xylan is among the most found hemicelluloses, and its backbone comprises β-1,4-xylose units in a wide range of conformations, with substitutions by arabinose (Ara) or glucuronic acid (GlcA), and modifications by acetyl (Ac) groups. Lignin contains guaiacyl (G), syringyl (S), and p-hydroxyphenyl (H) phenolic residues, which are interconnected by different types of covalent linkers such as β-O-4 ether-O-aryl, β-β' resinol, and β-5' phenylcoumaran[14,15].

Conceptually, the mechanical scaffold of crystalline cellulose is dispersed in a matrix formed by hemicellulose and lignin[16]. Our understanding of cell wall organization is supported by many studies that employed diffraction methods to reveal the spatial arrangement of cellulose microfibrils, imaging techniques to map out cell wall meshes and the microscopic distribution of lignin, and solution nuclear magnetic resonance (NMR) spectroscopy to identify lignin–carbohydrate linkages[17–19]. However, the interface between lignin and polysaccharides, the focus of this study, is not yet well understood. This is partially due to the hardly accessible length scale (angstrom to nanometer) and the requirement of both chemical and atomic resolutions. In addition, only a small number of molecules reside on this lignin–carbohydrate interface, which needs to be deconvoluted from the bulk of the cell wall. As both lignin and polysaccharide exist in the solid state, conventional separation methods often perturb their structures and interactions, making it difficult to investigate this polymer interface.

Recently, multidimensional solid-state NMR spectroscopy of Arabidopsis and Zea mays (maize) has spotlighted a structure-function relationship of the molecules involved in the lignocellulosic interface[20,21]. Lignin tends to form hydrophobic and disordered nanodomains, the surface of which binds the xylan in a three-fold helical screw conformation (three sugar residues per helical turn: a non-flat structure; Fig. 1a) through noncovalent interactions. The three-fold domain is connected to its two-fold flat-ribbon region, which is coating the smooth surface of cellulose microfibrils[22–25]. To generalize these structural principles, we need to examine other plant systems to evaluate three critical aspects: (i) the conformational bias of hemicellulose's function,

(ii) the absence of cellulose–lignin contact, and (iii) the self-aggregating nature of aromatic polymers.

The combination of solid-state NMR and dynamic nuclear polarization (DNP) methods has allowed us to unveil the structural and chemical principles underlying the formation of lignocellulosic materials. We investigated the $^{13}$C-labeled stems of two hardwoods, eucalyptus (Eucalyptus grandis) and poplar (Populus × canadensis), and the softwood spruce (Picea abies). These plants are non-food energy candidates for the development of second-generation biofuels to reduce our dependence on grain crops[26]. Despite lignin's preference for binding non-flat xylan, direct contacts are also observed between the aromatics and the junctions of cellulose surface and flat-ribbon xylan, which is probably enforced by molecular crowding. In these woody plants, molecules experience a highly homogeneous mixing on the nanoscale, which inevitably involves polymer interpenetration and entanglement. Consequently, the counter-intuitive hydration of aromatics is observed, where water is retained and stabilized by the polysaccharides closely packed to lignin. These molecular insights have brought our understanding of lignocellulose architecture to an unprecedented level of detail, allowing us to envision better biomass-conversion schemes for sustainable energy production.

## Results

**Polymorphic structure of carbohydrates in woody stems.** We obtained $^{13}$C-labeled plant stems by providing poplar, eucalyptus (gum tree), and spruce with $^{13}$CO$_2$. Free from chemical treatments, the hydrated lignocellulosic materials were directly analyzed using solid-state NMR. Therefore, all biomolecules have fully retained their chemical structure and physical packing. Atomic-level information of polysaccharide structure was obtained using two-dimensional (2D) $^{13}$C–$^{13}$C correlation experiments (Supplementary Fig. 1), which spotted 69 allomorphs of monosaccharide residues in three samples (Supplementary Data 1 and Supplementary Table 1). These sugar units, with a wide range of linkages and conformations, were mainly found in cellulose and four hemicelluloses, including glucuronoxylan, arabinoxylan, galactoglucomannan, and a very low amount of xyloglucan (Fig. 1a).

The remarkable resolution is evidenced by the narrow $^{13}$C linewidths of 0.5–0.9 ppm, which allowed us to inspect the multifaceted polymorphism of polysaccharide structures. Locally, the structural variation of cellulose happens to the glucose hydroxymethyl conformations defined by the O5 − C5 − C6 − O6 (χ) torsional angle (Fig. 1a)[27]. The surface (s) and internal (i) chains primarily and, respectively, adopt gauche-trans (gt, χ = +60°) and trans-gauche (tg, χ = 180°) conformations[27], which resulted in well-resolved signals (Fig. 1b). Upon bundling, the averaged structure of a fibril was restrained using the number of glucan chains residing in different environments, including the hydrophilic (s$^f$) or hydrophobic (s$^g$) surface, the inaccessible core (i$^c$), and the middle layer (i$^{a,b}$) sandwiched in between (Fig. 1c)[28]. A satisfactory fit includes three elementary microfibrils, with 15, 12, 6, and 21 chains for the s$^f$, s$^g$, i$^c$, and i$^{a,b}$ forms, respectively. Assuming uniform interfibrillar association, this averaged structure only designates the minimal bundle size, without considering loose packing. Other arrangements disagree with spectral observables (Supplementary Fig. 2).

Hemicellulose structure is highly complex as evidenced by the peak multiplicity. The backbones of hardwood xylan encompass two-fold (Xn$^{2f}$), three-fold (Xn$^{3f}$), and mixed (Xn) conformations (Fig. 1d). The mixed form is absent in spruce; therefore, softwood xylan has higher homogeneity. Xylan was found to be rigid, suggestive of a close association with the stiff cellulose

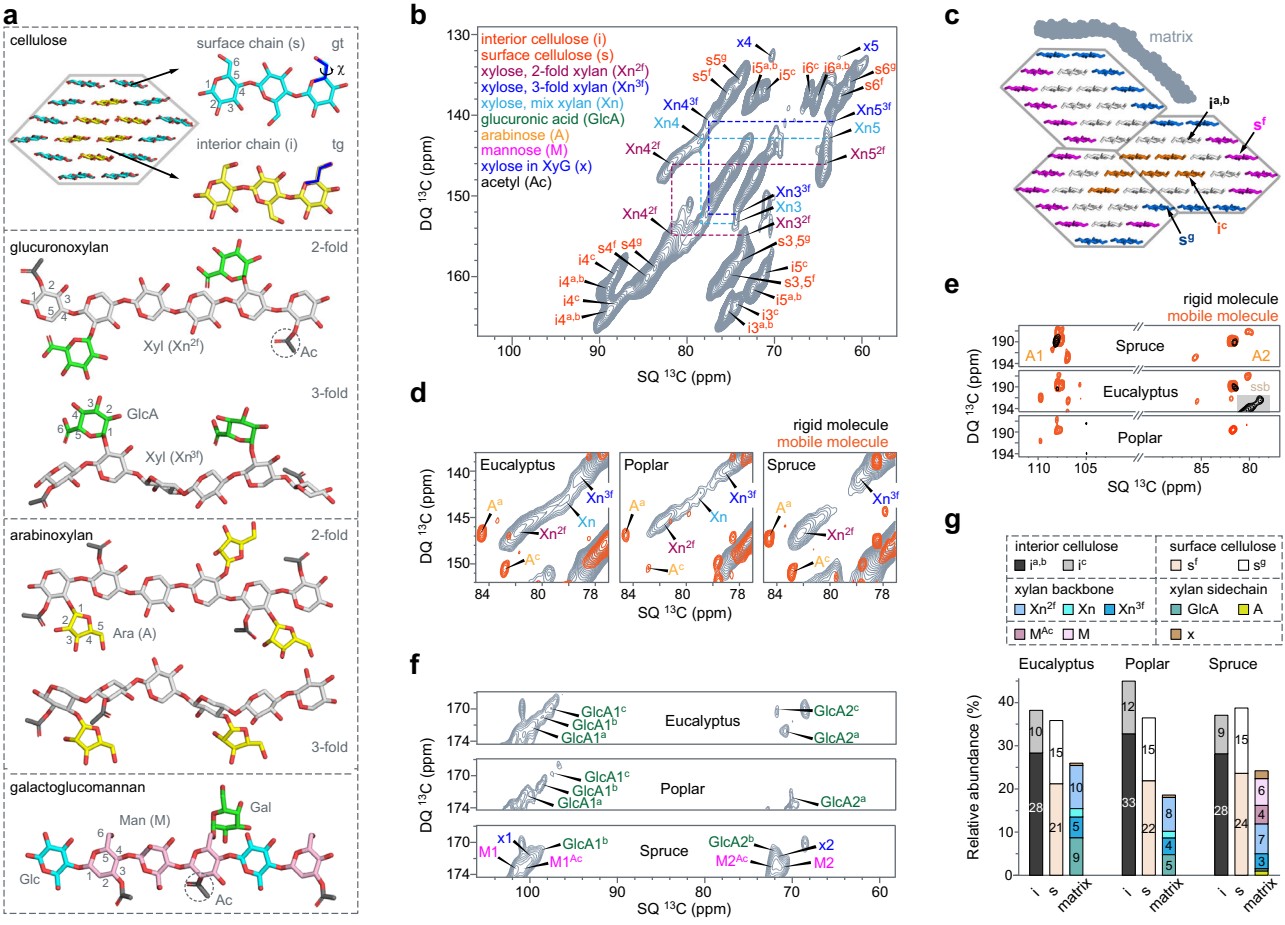

**Fig. 1 Solid-state NMR resolves the polymorphic structure of polysaccharides using intact wood cells. a** Representative structures of cellulose and major types of hemicellulose (xylan and galactoglucomannan). The cross-section of an elementary cellulose microfibril is shown, with a close view of an individual glucan chain and the hydroxymethyl conformation of surface (gauche-trans, gt) and internal (trans-gauche, tg) glucan chains. Xylan has two-fold or three-fold helical screw conformation in the backbone ($Xn^{2f}$ and $Xn^{3f}$), with glucuronic acid (GlcA) or arabinose (Ara or A) sidechains. Acetyl motifs (Ac) are present in hemicelluloses. **b** Representative 2D $^{13}C$ J-INADEQUATE spectrum of eucalyptus, which is based on $^{13}C$ cross polarization selecting rigid molecules. Dash lines track the carbon connectivity of xylan conformers. The NMR abbreviations of the carbohydrates are given. **c** A model of three elementary cellulose microfibrils fitting the NMR observables. The hydrophilic ($s^f$) and hydrophobic ($s^g$) surfaces, the embedded chains ($i^c$), and the middle layer ($i^{a,b}$) are color-coded. **d** Carbon-4 regions measured using $^{13}C$ direct polarization (orange spectra selecting mobile molecules) and cross polarization (gray spectra selecting rigid molecules). Both two- and three-fold xylan backbones are rigid while arabinose is mobile. **e** Arabinose carbon-1 and carbon-2 regions. The only rigid arabinose is the terminal one in xylan sidechain. **f** Eucalyptus has a high content of GlcA and spruce is rich in acetylated or non-acetylated mannose ($M^{Ac}$ or M). **g** Composition of the rigid molecules in cell walls determined by peak volumes from 2D spectra. Source data are provided as a Source Data file.

microfibrils. This observation differs from previous findings in *Arabidopsis*, where three-fold xylan remained mobile due to its spatial separation from cellulose[23]. Four types of arabinose (A) signals were identified, likely induced by the varied linkage sites at carbon 1, 2, 3, and 5 (Fig. 1e). Only a single type is partially rigid, which is attributed to the terminal arabinose of xylan sidechains in secondary cell walls, whereas the mobile ones are from pectic polymers in primary walls. Hardwoods showed a high content of glucuronic acid (GlcA), while the softwood spruce exhibited unique signals of mannoses (M), one of the major constituents of galactoglucomannan (Fig. 1f)[29].

Analysis of the spectral intensities led to the molar composition of rigid polysaccharides (Fig. 1g). Around three quarters (74–81%) are cellulose, making it the most abundant polymer in woody stems (Supplementary Table 2). Xylan makes up 18–26% of hardwood polysaccharides. Spruce has equal shares of xylan and mannan, each accounting for 10–12%. Consistent across these samples, the amount of two-fold xylan has doubled that of the three-fold counterpart, likely due to their promoted

interactions with cellulose. Xylan sidechains are predominantly GlcA (therefore, glucuronoxylan) in hardwoods but mainly Ara (that is, arabinoxylan) in spruce.

**Complex structure and linkage of lignin**. Wood lignin mainly contains guaiacyl (G) and syringyl (S) residues, with a single and two methoxyl groups, respectively (Fig. 2a, b). The unsubstituted ring, *p*-hydroxyphenyl (H), was not observed in solid-state NMR spectra due to its low abundance in these plants. The aromatic signals are dispersed over an extensive range of chemical shifts. For instance, four types of S/S′ residues and four forms of G units were identified in eucalyptus and spruce, respectively. The multiplicity should be triggered by the varied oxidation states (for example, S′ is a Cα-oxidized form of S unit), the assorted linkages, and presumably, the wide-ranging conformation and packing in native solids.

Aromatic rings are interconnected by a diverse array of covalent linkers. For example, β-ethers (A), with a characteristic

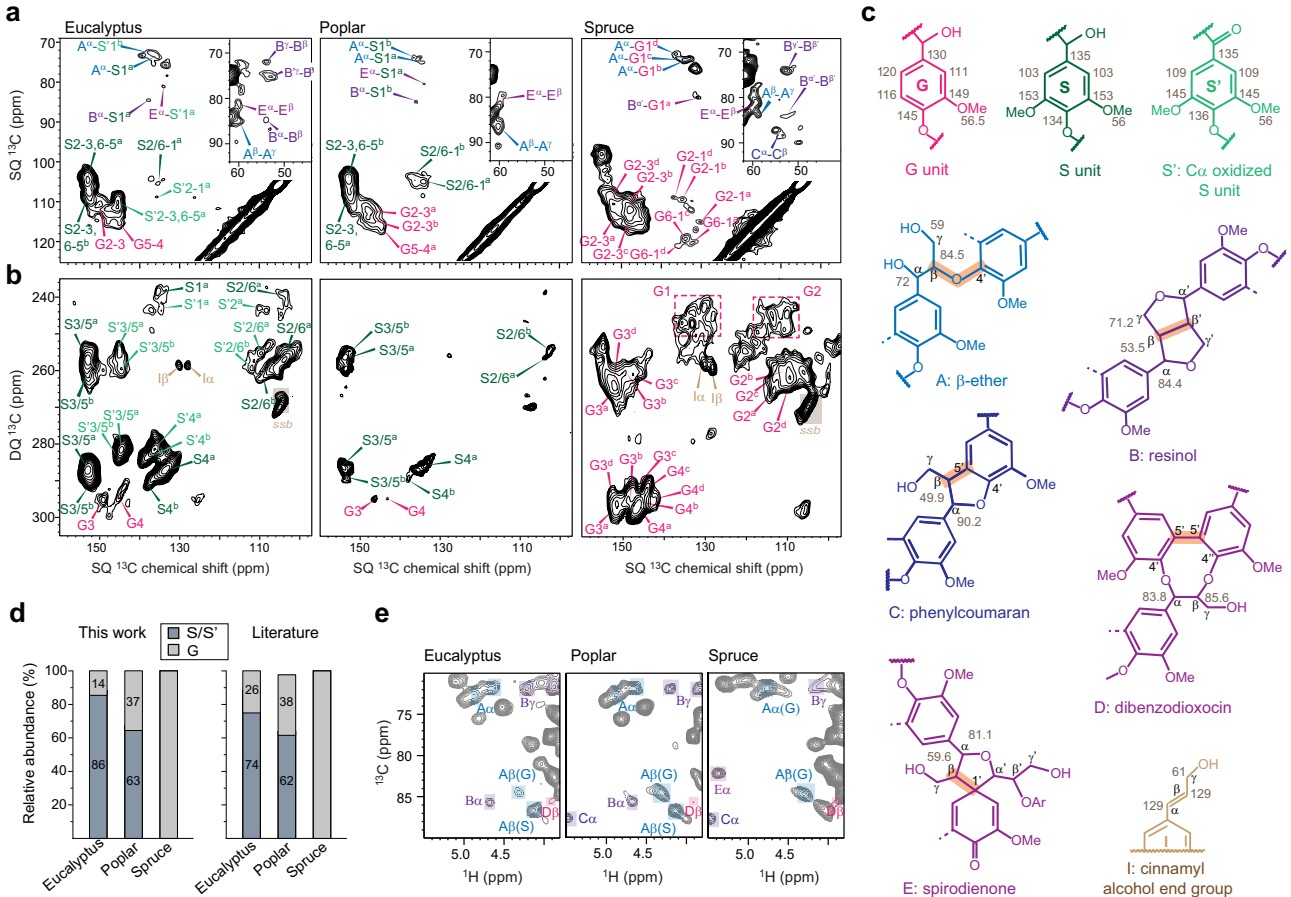

**Fig. 2 Chemical structure of lignin determined using solid-state and liquid NMR.** Lignin signals are resolved using the aromatic regions of **a** 2D $^{13}$C-$^{13}$C RFDR spectra and **b** 2D $^{13}$C J-INADEQUATE spectra of intact plant cells of eucalyptus, poplar, and spruce. **c** The structure of lignin units and linkages resolved in wood stems. The standard abbreviations are given under each structure. The observed $^{13}$C chemical shifts are labeled for each carbon site. Key inter-residual linkages are highlighted. **d** Composition of lignin units quantified using the integrals of one-bond cross peaks in 2D $^{13}$C RFDR spectra (left panel) and using solution NMR as reported in the literature (right panel)[30,32,33,35,73,85,86]. Eucalyptus and poplar are S-rich while spruce is G-rich. Source data are provided as a Source Data file. **e** Representative lignin linkages detected by 2D $^{1}$H-$^{13}$C HSQC spectra of ball-milled biomass in d$_6$-DMSO.

one-bond cross peak between carbons β and γ at (85, 59) ppm (Fig. 2a), have the β-O-4 linkages that are readily cleavable during degradation. Other signals emerged from resinol (B) that has β-β connections, phenylcoumaran (C) that encompasses β-5 and α-O-4 linkages and spirodienone (E) that contains β-1 and α-O-α links (Fig. 2c). These signals are detectable by two experimental schemes based on $^{1}$H–$^{13}$C cross polarization (CP) and $^{13}$C direct polarization (DP) methods, indicating the distribution of these linkers in the rigid and mobile phases of lignin (Supplementary Fig. 3). Despite the low intensity, we managed to detect several cross peaks between these linkers and aromatic carbons (Fig. 2a). For example, the connection between the carbon α of β-ethers and the carbon 1 of S residues yielded a cross peak at (72, 135) ppm in eucalyptus and poplar. Similar correlations were also found between the carbon 1 of S residues and the carbon α of B and E linkers. In spruce, analogous connections were observed for G units. These junctions typically evade solution-NMR characterization because the aromatic carbon 1 of lignin is nonprotonated. These results demonstrated the feasibility of using solid-state NMR to characterize lignin structure and linkage, but the technical capability still requires further development.

Hardwood samples are rich in S residues (63–86 mol%) while the softwood only contains G units (Fig. 2d). This observation agrees with our freshly collected solution-NMR data of ball-milled and dissolved lignocellulose (Supplementary Fig. 4 and Table 3), average S/G ratios found in the literature (Supplementary Table 4)

as well as the quantification using deconvoluted 1D solid-state NMR spectra (Supplementary Fig. 5 and Table 5). Both solid-state and solution-NMR results (Fig. 2e) suggest that these woody plants contain a large amount of β-ethers[30]. Solution-NMR spectra also show considerable signals of both resinol and phenylcoumaran in poplar and spruce (Fig. 2e), while the phenylcoumaran peaks become very weak in eucalyptus[15,30–36]. Since these linkers respond differently to degradation, such analysis helps identify the plant candidates for saccharification.

**Distinct patterns of lignin–carbohydrate packing across plant species.** The supramolecular architecture of lignocellulose differs dramatically in hardwood and softwood. In both eucalyptus and poplar, the use of a long-mixing period (1.0 s) in 2D $^{13}$C–$^{13}$C correlation spectra has generated many long-range intermolecular cross peaks that are absent in the short-mixing (0.1 s) spectrum (Fig. 3a and Supplementary Fig. 6). Puzzlingly, these two spectra showed a comparable pattern in spruce, which signified that $^{13}$C magnetization was already equilibrated among polymers within 0.1 s. This is not caused by variations of spin diffusion coefficients or polymer dynamics in hardwood and softwood as validated experimentally (Supplementary Figs. 7 and 8). Therefore, the rapid equilibrium observed in spruce indicates that lignin and polysaccharides are well-mixed on the nanoscale in spruce but stay apart in hardwoods.

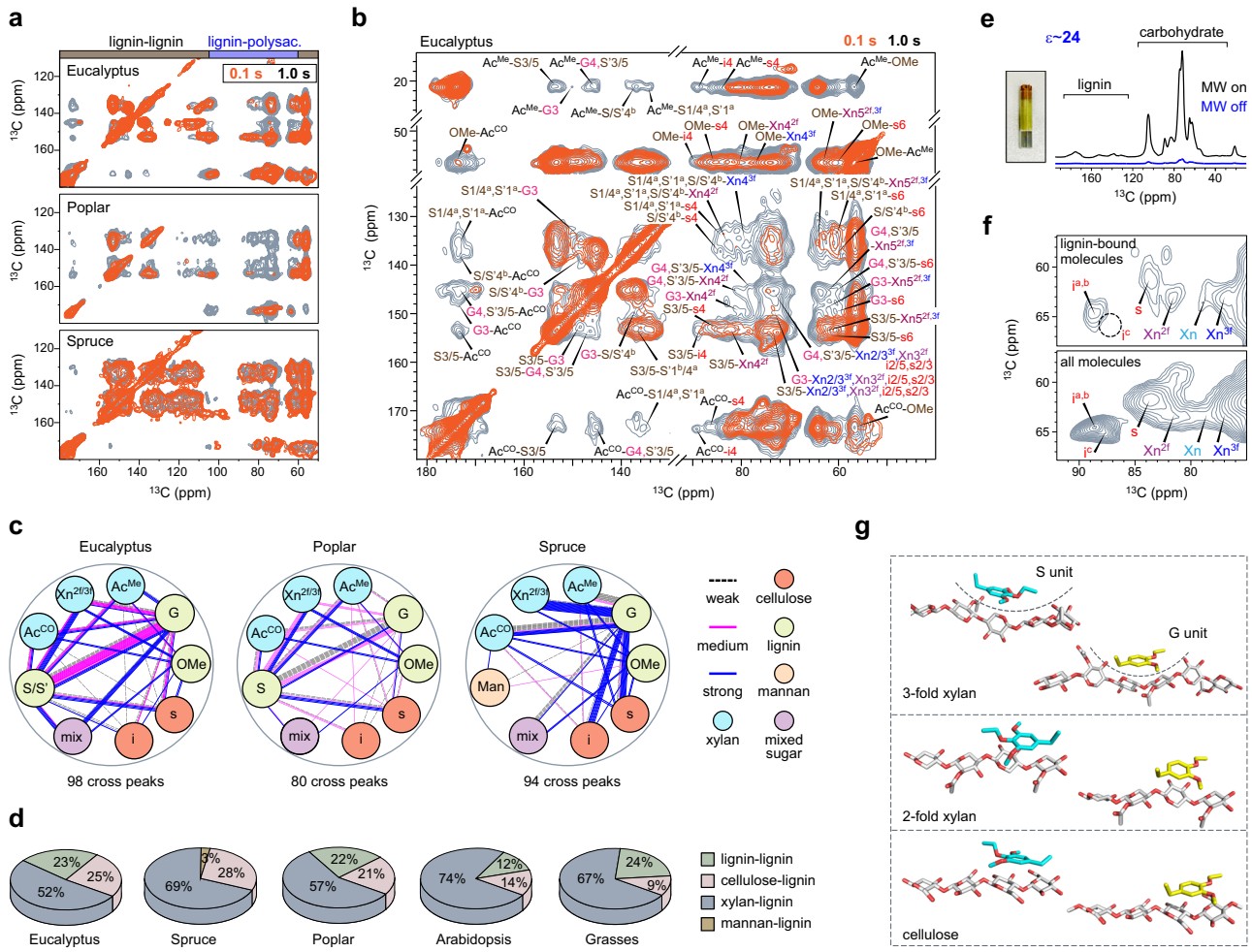

**Fig. 3 DNP-assisted detection of long-range interactions defines lignin–carbohydrate packing. a** Overlay of long-range (1.0 s mixing; grey) and short-range (0.1 s mixing; orange) 2D $^{13}$C-$^{13}$C correlation spectra. The long- and short-range spectra showed similar spectral patterns in spruce: softwood polymers are homogeneously mixed on the molecular level. **b** Dipolar-gated 2D $^{13}$C-$^{13}$C correlation spectra showing 98 intermolecular cross peaks in eucalyptus. **c** Summary of 272 intermolecular interactions identified in eucalyptus, poplar, and spruce. Cellulose, lignin, mannan, and mixed sugars are color-coded in red orange, light green, light orange, and purplre, respectively. Source data are provided as a Source Data file. **d** The count of lignin–lignin, cellulose–lignin, and xylan–lignin interactions in eucalyptus, poplar, spruce, *Arabidopsis*, and grasses (maize, rice, and switchgrass). Lignin–cellulose interactions are scarce in *Arabidopsis* and grasses but become abundant in woods. Source data are provided as a Source Data file. **e** DNP enhancing NMR sensitivity of eucalyptus by 24-fold. Inset shows a picture of the plant material in a sapphire DNP-NMR rotor. **f** Comparison of the equilibrium spectrum (bottom) of eucalyptus that detects all components with the aromatic-edited spectrum (top) that only shows lignin-bound molecules. The lignin-bound polysaccharides include the two-fold, three-fold, and mixed conformers of xylan as well as the surface and interior cellulose. The deeply embedded glucan chains in cellulose ($i^c$) are absent as highlighted using the dash line circle. **g** DFT energy-minimized structures showing the possible packing between lignin units and polysaccharides.

The spatial association of polymers was pinpointed by 272 intermolecular cross peaks, each of which represents a sub-nanometer, physical contact between two carbons from different molecules (Fig. 3b and Supplementary Fig. 9). For example, the methyl carbon in the acetyl group of xylan (Ac$^{Me}$) exhibited cross peaks with the carbon 3 or 5 of S lignin (S3/5) at (21, 153) ppm and with the carbon 3 of G-lignin (G3) at (21, 148) ppm (Fig. 3b). These interactions happen between (i) acetyl groups and lignin, (ii) xylan and lignin, (iii) acetyl and cellulose, (iv) acetyl and lignin methyl, (v) lignin methoxyl group and cellulose, and (vi) different lignin residues (Supplementary Fig. 10)[21].

A statistical view of the number and intensities of intermolecular contacts (Supplementary Table 6 and Supplementary Data 2–4) designated xylan as the major interactor with lignin, which is manifested by the extensive correlations between G/S and xylan carbons, including the carbonyl and methyl carbons of acetyl (Ac$^{CO}$ and Ac$^{Me}$) and xylose ring carbons (Fig. 3c). This

result echoes with the recent findings in *Arabidopsis* and commelinid monocots (grasses, for example, maize and switchgrass)[24], validating the principal role of xylan–lignin interaction in stabilizing lignocellulose. Mannan has a small number of resolvable sites; therefore, it only showed a few cross peaks with lignin. However, the equilibrated pattern in Fig. 3a supports a widespread association of mannan and lignin in spruce.

Unexpectedly, wood lignin also exhibited extensive correlations with cellulose (Fig. 3c), which accounts for 20–30% of all intermolecular cross peaks (Fig. 3d). Such interactions are particularly abundant in spruce, consistent with the homogeneous mixing of molecules in this sample. In contrast, cellulose–lignin contacts only account for 9–14% of polymer interactions in *Arabidopsis* and grasses[24]. Thus, cellulose–lignin interaction is a unique feature of woody plants, which should contribute, at least in part, to the mechanical strengths of their stiff stems.

**Visualization of the polysaccharide–lignin interface**. The unanticipated lignin–cellulose interactions were verified using the sensitivity-enhancing DNP technique[37–39]. The eucalyptus stem showed a 24-fold enhancement of NMR intensity (Fig. 3e), which was achieved by transferring the polarization of electrons in the stable biradical AMUPol to the protons of biopolymers. This sensitivity enhancement shortens the measurements by 576 times (a 1.5-year experiment in a day), allowing us to take snapshots of the lowly populated boundaries between polysaccharides and lignin. Eucalyptus cellulose was found in the lignin-bound part of polysaccharides (Fig. 3f), but such signals were missing in maize[24]. Only the surface glucan chains of cellulose (s) or those chains right underneath the surface layer (i$^{a,b}$) correlated with lignin, while the embedded core (i$^c$) lacked such interaction. The three-fold xylan (Xn$^{3f}$) is still favorable for binding lignin, showing stronger signals than the two-fold (Xn$^{2f}$) and the mixed (Xn) conformers.

Favorable lignin–polysaccharide interactions were revealed using quantum mechanical geometry optimizations conducted using the DFT method in continuum solvation models (Fig. 3g). The three-fold xylan formed a packet to enclose aromatics. Conversely, the flat chains from cellulose surface or two-fold xylan relied on their pyranose/furanose rings to align with an aromatic unit. Examination of recently reported DFT structures[40] showed that S-units preferentially aligned to carbohydrate rings as stabilized by the two methoxyl groups on both sides of the aromatic ring (Supplementary Fig. 11). The G unit, on the other hand, typically has its single methoxyl group closer to the carbohydrates. Such orientational preference supports the strong interactions experimentally observed between lignin methoxyl groups and xylan acetyls (Supplementary Fig. 10). Consequently, the methoxyl-rich S-residues correlate better with xylan in space. Therefore, noncovalent interactions between these polar groups are essential to the existence of lignin–xylan complex.

**Landscape of biomolecular hydration and dynamics**. Biopolymers have sophisticated dynamics and variable water association in their native environments. Among the three plants, poplar turned out to be the worst hydrated sample as shown by its slowest water-to-polymer $^1$H polarization transfer buildup curves, which were consistent for both cellulose and xylan (i4 and Xn$^{3f}$1, Fig. 4a), as well as lignin (Supplementary Fig. 12). The relative intensities (S/S$_0$) of a hydration heatmap reflect the degree of water association (Fig. 4b and Supplementary Fig. 13). Plots of the S/S$_0$ ratios against 127 carbon sites show that the hydration level increases sequentially from poplar to eucalyptus and spruce (Fig. 4c). Within each sample, polymer hydration generally increases from cellulose to xylan, and then to mannan, if present.

Surprisingly, wood lignin retained high S/S$_0$ ratios, typically on the same range as those of xylan, indicating a high level of hydration (Fig. 4c). This is intriguing because these aromatic polymers are perceived as relatively hydrophobic, which by expectation, should hinder water retention. For example, maize lignin was previously found to self-aggregate, forming nanodomains that are largely separated from water[24]. The water contact observed here should originate from lignin's tight association with those carbohydrates that kept water localized. Likewise, wood biopolymers should experience frequent entanglement and interpenetration instead of domain separation.

Polymer dynamics were examined using $^{13}$C–T$_1$ and $^1$H–T$_{1\rho}$ relaxation measurements, which generated 150 relaxation curves (Supplementary Figs. 14 and 15). When all molecules were considered, the $^{13}$C–T$_1$ relaxation time decreased in the order of cellulose, lignin, xylan, and mannan, if any (Fig. 4d). The short $^{13}$C–T$_1$ time constants of lignin and hemicellulose revealed the

efficient $^{13}$C–T$_1$ relaxation in these non-cellulosic polymers, and furthermore, their enhanced motion on the nanosecond timescale. Such differences became indistinguishable if only rigid molecules were focused on, for example, all rigid molecules of eucalyptus showed $^{13}$C–T$_1$ time constants of 4–5 s. This observation contradicts previous results in which maize lignin showed substantially longer $^{13}$C–T$_1$ than any polysaccharide[24]. This can be comprehended using the efficient spin-exchange between lignin and carbohydrates (mediated by the better molecular mixing), which has averaged the $^{13}$C–T$_1$ relaxation times in wood. On the microsecond timescale, cellulose had the longest $^1$H–T$_{1\rho}$ relaxation times (30–40 ms) due to the restricted motion in the massively hydrogen-bonded microfibrils (Fig. 4e). Both lignin and hemicellulose had short $^1$H–T$_{1\rho}$ times of 10–15 ms, like the counterparts in maize[24]. Compared to G residues, S units exhibited slower $^1$H–T$_{1\rho}$ relaxation, indicating attenuated dynamics due to interactions with polysaccharides (Supplementary Fig. 15). Among the three samples, spruce displayed the shortest $^{13}$C–T$_1$ and $^1$H–T$_{1\rho}$ relaxation times, revealing a unique profile of molecular dynamics in softwood.

**Effects of rehydration on water retention and polymer dynamics**. Water molecules are important for stabilizing the nanostructure of cell wall biomaterials, and the complete removal of water could potentially cause irreversible changes. In a recent NMR study of the softwood *Pinus radiata*, oven-drying (for complete dehydration) and rehydration were found to promote polymer association (e.g., xylan–cellulose and lignin–cellulose packing) and irreversibly alter the dynamics and conformation of mannan in the secondary cell walls[41]. Similarly, the lyophilized-rehydrated grass sorghum (*Sorghum bicolor* L. Moench) has shown permanent changes of hemicellulose, with enhanced mobility for the three-fold xylan backbone and some arabinose residues[42]. Differently, dehydration only caused reversible structural changes to the primary cell walls of *Arabidopsis*, likely due to the high content of pectic polymers that can efficiently associate with water molecules[43].

We measured the never-dried samples of spruce and eucalyptus, followed by re-examination of the same materials after lyophilization and rehydration. In spruce, the lignocellulosic complex was not dramatically affected by the rehydration process, which differs from lipids and proteins (Fig. 5a). The structure of the carbohydrates and lignin, as well as their distribution in dynamically distinct domains, can be efficiently retained in the rehydrated biomass, occasionally with minor intensity variations that are well below 10% for each carbohydrate or lignin carbon site (Fig. 5a). The retention of water molecules can also be fully restored, as shown by the identical spectral patterns in 1D water-edited spectra (Fig. 5b) and the comparable 2D heatmap representation for both carbohydrates and aromatics in the never-dried and rehydrated samples (Fig. 5c). The nanosecond-timescale motion and the subnanometer polymer packing (reflected by spin-exchange) are unaffected since there is no deviation of $^{13}$C–T$_1$ relaxation within the well-controlled error margin (Fig. 5d). Both lignin and carbohydrates showed a 10–20% increase in the $^1$H–T$_{1\rho}$ relaxation time constants; therefore, the only notable change happened to the microsecond scale motions of biopolymers (Fig. 5e). Eucalyptus behaves differently: this plant fully restored the structural and dynamical features but moderately improved polymer-water association after rehydration (Supplementary Fig. 16).

**Discussion**

The abundant molecular-level evidence presented four novel features of lignocellulosic materials, each exploring a structural or

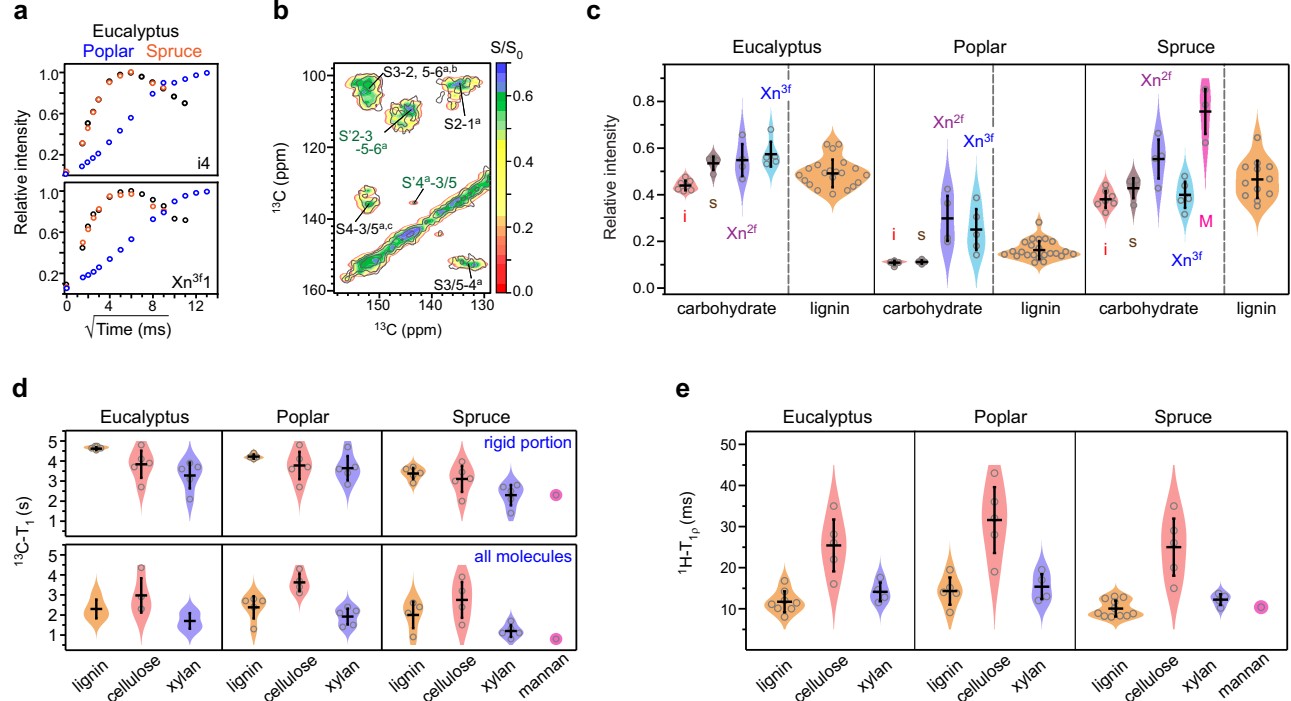

**Fig. 4 Site-specific hydration and dynamic landscape of biomolecules. a** Representative water-to-biomolecule $^1$H polarization transfer buildup curves. The data of interior cellulose carbon 4 (i4) and three-fold xylan carbon 1 (Xn$^{3f}$1) are compared across plants. Poplar has slow buildup due to the limited water contact. **b** Hydration map on top of a 2D spectrum plotting the ratio ($S/S_0$) of water-edited intensity (S) to the equilibrium intensity ($S_0$). A larger $S/S_0$ ratio indicates better polymer hydration. **c** Distribution of the relative water-edited intensities ($S/S_0$) of polysaccharides and lignin in eucalyptus, poplar, and spruce. Molecules with better water retention show higher water-edited intensities. Poplar is poorly hydrated. Relative intensities: $n = 7, 4, 4$, and 5 for i, s, Xn$^{2f}$, and Xn$^{3f}$, respectively. For lignin, $n = 21, 23$, and 12 for eucalyptus, poplar and spruce, respectively. $n = 3$ for the mannan in spruce. Source data are provided as a Source Data file. **d** $^{13}$C–$T_1$ relaxation times of rigid (blue rectangles) and all molecules (colored as yellow, red, purple, and magenta for different molecules) in three woody plants, which represent nanosecond-timescale motions. $^{13}$C–$T_1$ relaxation time constants: $n = 5$ for both cellulose and xylan in all three plants. For lignin, $n = 5, 4$, and 5 for eucalyptus, poplar and spruce, respectively. Source data are provided as a Source Data file. **e** $^1$H–$T_{1\rho}$ relaxation times reflecting microsecond timescale dynamics. $^1$H–$T_{1\rho}$ time constants: $n = 5$ for cellulose and $n = 4$ for xylan in all plants; $n = 8, 5$, and 11 for lignin in eucalyptus, poplar and spruce, respectively. The average value and standard deviation (error bars) are presented for each violin plot in panels **c–e**, the dataset of which are tabulated in Supplementary Tables 7–11. Source data are provided as a Source Data file.

chemical foundation of the supramolecular architecture (Fig. 6). First, even though three-fold xylan is favored for binding lignin, in wood, other xylan conformers can also coexist with lignin in part. Accompanying with this functional resemblance is the hydration equivalence of these xylan forms (Fig. 4c). These results have extended the conceptual model of lignocellulose derived from maize, in which lignin mainly interacts with the three-fold xylan, and vice versa. Actually, interactions between three-fold xylan and cellulose have been reported in a grass *Sorghum*[42], which also implies the interchangeable roles of xylan conformers. Uniquely, wood xylan is mainly in two-fold helical screw, which is energetically unfavorable unless forced by the binding to cellulose surface. Therefore, the unanticipated proximity of lignin to the two-fold xylan, as well as its associated cellulose surface, might be a destined consequence of spatial crowding in densely packed lignocellulosic materials.

Second, the packing between cellulose and lignin is plant-specific and only serves as a secondary interaction. In *Arabidopsis* and grasses, lignin and cellulose are spatially separated[24]. The situation has changed for woody biomass, where lignin and cellulose become colocalized. Although *Arabidopsis* is widely used as a model system for investigating the biophysical traits of hardwoods, it becomes apparent that these plants differ in their cell wall organization. Our DFT results and a recent molecular simulation study[44] consistently suggest that lignin is mainly docked on the hydrophobic surface of cellulose, with aligned phenyl and pyranose rings. The increased coverage of cellulose surface by aromatic polymers might reduce enzyme accessibility and contribute to biomass recalcitrance.

Third, the structural feature of aromatic polymer needs reconsideration. Lignin nanodomains observed in maize are absent in woody plants due to the promoted polymer mixing on the nanoscale (Fig. 3). Accordingly, polymer entanglements and interpenetration, rather than superficial contact between domain surfaces, should govern lignin–carbohydrate interactions in wood[45]. Once well-mixed with polysaccharides, even the aromatics could effectively retain and immobilize water molecules (Fig. 4c). The thermodynamical driving force of this peculiar phenomenon awaits inputs from modeling methods.

Finally, the ultrastructure of softwood is featured by homogeneous molecular mixing (Fig. 3a). This structural hallmark has revised the prevailing model in which well-aligned and partially crystalline glucomannans were perceived as being sandwiched between an internal cylinder of cellulose microfibrils and an outside tubular domain of lignin, with xylan forming an outermost layer[46]. Actually, our results better align with a recent study reporting that glucomannan and xylan could potentially associate with both cellulose and lignin[29].

Although the focus of this exploratory study is to understand the chemical and physical principles underlying polymer packing and lignocellulose architecture, the structural foundation and methodology established here would inspire more in-depth

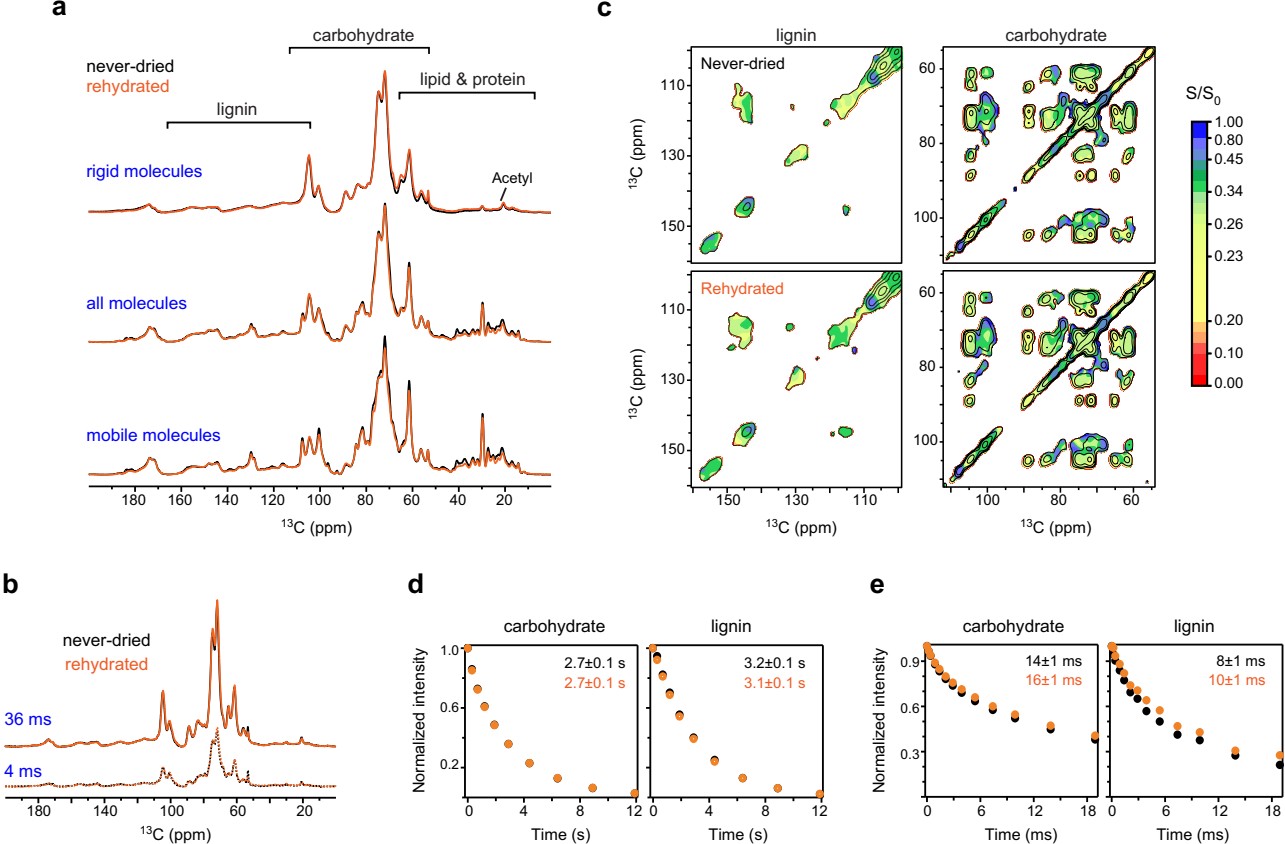

**Fig. 5 Molecular-level dynamics and water association after dehydration and rehydration.** A never-dried spruce sample was first measured, and then freeze-dried and rehydrated for direct comparison. **a** Overlay of spectra collected using never-dried (black) and lyophilized and then rehydrated (orange) spruce stem samples showing high similarity for polysaccharides and lignin. From top to bottom are the CP, quantitative DP, and short recycle-delay DP spectra. The key regions of lignin, carbohydrate, and lipid are marked. **b** 1D water-to-carbohydrate polarization transfer spectra revealing similar water association for never-dried and rehydrated samples. The well-hydrated molecules are selected using a short $^1$H mixing time of 4 ms while the equilibrated state is measured using 36 ms $^1$H mixing. **c** Water-edited 2D hydration map showing comparable water-edited intensities ($S/S_0$) for both lignin and carbohydrate regions in the never-dried and the rehydrated samples. The ratio ($S/S_0$) is between the water-edited intensity ($S$) and the equilibrium intensity ($S_0$). **d** Average $^{13}$C–$T_1$ relaxation plot, using the integrals of spectral regions for carbohydrates (left) and lignin (right). Source data are provided as a Source Data file. **e** Average $T_{1\rho}$ relaxation of carbohydrates and lignin in never-dried (black) and rehydrated (orange) samples. Error bars are standard deviations of the fitting derived parameters. Source data are provided as a Source Data file.

investigations for understanding the structural diversity and mechanical properties presented by numerous plant species and mutants, various cell types, and different growth stages[47]. These structural insights will guide the utilization of forestry resources for the production of biomaterials and biofuels[48], and the spectroscopic toolbox will stimulate structural investigations of polymer assemblies in other organisms, such as bacteria, fungi, and algae, as well as bio-inspired materials[49–58].

## Methods

**Plant material**. Uniformly $^{13}$C-enriched stems (97% $^{13}$C) of eucalyptus (*Eucalyptus grandis*; age 16 weeks), poplar wood (*Populus × canadensis*; 27 weeks), and spruce (*Picea abies*; 16 weeks) were obtained for structural characterization from IsoLife (The Netherlands). Eucalyptus and spruce were obtained from seeds while poplar was obtained from stem cutting. Poplar plants were saplings of close to 1 m height at harvest. Immediately after removing the plants from the growth chamber, plant shoots were dissected into leaves and stems. The stem-sections were split, cut, and debarked after freeze-drying. Debarking was strived to be complete, which was conducted through the longitudinal cutting of the bark followed by separation from the xylem at the cambium interface. Debarked stems were hydrated for NMR analysis. The material was briefly hand-grinded using a pestle and a mortar, resulting in small pieces on the range of a few mm across. The homogeneity helps to avoid potential issues during magic-angle spinning.

To test the effect of freeze-drying, we obtained never-dried $^{13}$C-labeled spruce and eucalyptus wood from IsoLife, without lyophilization. The debarking process is finished manually by carefully removing the outside soft bark layer with tweezers.

Around 104 mg of debarked spruce was cut into mm scale pieces, packed into 4-mm rotor, and measured under 10 kHz MAS on a 400 MHz Bruker NMR spectrometer. The never-dried samples were first measured freshly. After ssNMR measurements, overnight lyophilization was conducted, giving 43.5 mg of freeze-dried spruce stem. The sample is then sufficiently rehydrated (with a final weight of 107 mg), and all ssNMR experiments were conducted again on the rehydrated material for direct comparison with never-dried samples (Fig. 5). Besides, never-dried (96 mg) eucalyptus stem, and the dehydrated and then rehydrated material (94 mg including 68 wt% $H_2O$) were also examined (Supplementary Fig. 16).

**Isotope labeling**. Plant samples from eucalyptus, poplar, and spruce were produced under identical growth conditions in custom-designed, air-tight, high-irradiance labeling chambers of the Experimental Soil Plant Atmosphere System at IsoLife (Wageningen, The Netherlands). Under regulated environmental conditions, these plants were grown hydroponically in a closed atmosphere containing 97 atom% $^{13}$CO$_2$ from germinated seed or rooted stem cutting till harvest: photosynthetic photon flux density (PPFD) 800 µmol m$^{-2}$ s$^{-1}$ (top of plants); CO$_2$ concentration (Day); 400 ppm (v/v); 15 h (eucalyptus), 21 h (poplar), and 16 h (spruce) day lengths; day/night temperature of 22/16 °C (eucalyptus), 22/15 °C (poplar), and 24/20 °C (spruce). The day/night relative humidity is 75/75% for eucalyptus, 70/80% for poplar, and 70/75% for spruce. Minerals and water were supplied as aerated modified Hoagland nutrient solutions with micronutrients and iron[1,2], maintaining nitrogen concentration between 25 and 200 mg/L; pH close to 5; EC between 0.4 and 0.7 mS/cm; 25% of total nitrogen was supplied as ammonium. Plant shoots were dissected into leaves and stems after removing the plants from the growth chamber. After freeze-drying, these plant stems were kept stored at −20 °C, in the dark with silica-gel drying bags. All plants were rehydrated for solid-state NMR analysis.

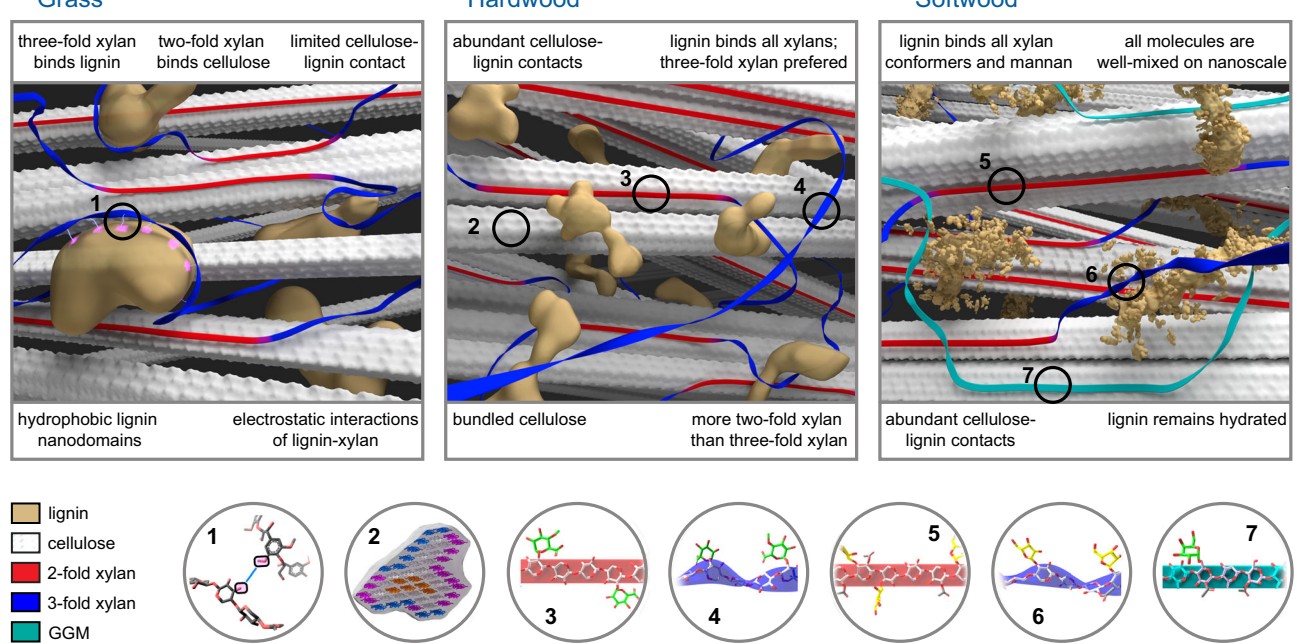

**Fig. 6 Comparative schemes of secondary cell walls in grass, hardwood, and softwood.** The figure shows the spatial arrangement of lignin (yellow), cellulose (white fibrils), two-fold xylan (red flat ribbons), three-fold xylan (blue twisted ribbons), mixed forms of xylan (magenta), and mannan (GGM; green) in secondary plant cell walls. Numbered spheres highlight the structural features of (1) lignin–xylan interaction in all plants, (2) cellulose bundles in woody plants, (3–6) two-fold and three-fold xylan with different sidechains, and (7) galactoglucomannan. The molecular fraction of polysaccharides is considered in the depiction, but the illustration may not be strictly to scale. The model of grass cell wall is generated using the data recently reported[24], for comparison with the models of woody plants. Lignin and carbohydrates are much better mixed in woody plants than in grass, resulting in the binding of lignin to both three-fold and two-fold xylan as well as to cellulose microfibrils. The structural assembly of woody cell walls is thus different from the domain-separation scheme of grass cell walls.

**Effects of plant age on cell walls**. Given our ever-increasing knowledge on plant secondary cell wall, impacts of the plant age at which the material is harvested relative to its natural growth cycle can be considered. Our 6.5-months old poplar can be put in perspective of 3-, 6- and 18-month-old poplar samples reported in literature[15]. Overall, there is no significant change in cellulose to hemicellulose proportions, and cellulose crystallinity has become a stable parameter after 6 months. However, a slight increase in lignin content with age has been consistently reported for both poplar and eucalyptus[30]. Solution-NMR data have demonstrated that lignin structure undergoes substantial evolution over a longer period of time, in both unit nature. For example, S-lignin is deposited at later stages and the proportions of different linkage types. In-depth analyses of polysaccharide structure, lignin composition and linkage types, the consequential effect on lignocellulose architecture is thus of significant interest for future solid-state NMR research.

**Solid-state NMR experiments for assignment**. The plant stems were packed into MAS rotors for measurements on 600 MHz and 400 MHz Bruker spectrometers using 3.2 mm and 4 mm MAS probes, respectively. Most experiments were collected under 10–14 kHz MAS at 293 K unless otherwise stated. $^{13}$C chemical shifts were referenced to the tetramethylsilane (TMS) scale. Radiofrequency field strengths were 80–100 kHz for $^1$H decoupling, 62.5 kHz for $^1$H CP contact pulse, and 50–62.5 kHz for $^{13}$C. The technical parameters of all solid-state NMR experiments were summarized in Supplementary Tables 12 and 13 for the initial batch of rehydrated samples, and in Supplementary Table 14 for the more recently obtained never-dried and the subsequent freeze-dried and rehydrated samples.

To assign NMR signals, 2D correlation spectra were recorded using the refocused CP J-INADEQUATE sequence[59], which was coupled with $^{13}$C-DP for detecting mobile molecules or $^1$H–$^{13}$C CP for detecting rigid polymers. A 2D $^{13}$C radio frequency-driven recoupling (RFDR) correlation experiment was measured under 10 kHz MAS and 83 kHz $^1$H decoupling to assign intraresidue cross peaks[60]. A recoupling time of 1.6 ms was used to detect one-bond $^{13}$C–$^{13}$C cross peaks. An additional set of $^{13}$C-DP PDSD experiments were conducted to examine the structure of lignin in the mobile phase, which was selected through the short recycle delay of 2 s. The standard flow chart of resonance assignment is presented in Supplementary Fig. 17. All chemical shifts are validated by comparison with literature-reported values (Supplementary Data 5).

To enhance aromatic signals, we measured a dipolar-coupling-gated version of the proton-driven spin diffusion (PDSD) experiment[24,61,62]. A dipolar-dephasing period of 48 μs was employed to efficiently suppress the signals of protonated

carbons, allowing the preferential detection of non-protonated carbons in the aromatic lignin. This period was asymmetric with respect to the π pulse in the Hahn echo, containing two undecoupled delays of 32 μs and 16 μs. A 0.1 s PDSD mixing was used to report intramolecular cross peaks in all samples. A 20 ms mixing was also used for spruce. The observed chemical shifts were compared with values deposited in the Complex Carbohydrate Magnetic Resonance Database to facilitate resonance assignment[63].

**Solid-state NMR experiments for structural analysis**. The 2D gated PDSD experiment was also conducted to determine lignin–carbohydrate packing, with a long-mixing time (1 s) for intermolecular polarization transfer. We have identified 98, 80, and 94 cross peaks for eucalyptus, poplar, and spruce, respectively. These 272 correlations were categorized as 112 strong, 75 medium, and 85 weak restraints according to their relative intensities (area of a peak relative to that of the 1D cross-section), with cutoffs set to >4.0% (strong), 2.0–4.0% (medium), and <2.0 (weak) as tabulated in Supplementary Data 2-4.

Polymer hydration was determined using water-edited 2D $^{13}$C–$^{13}$C correlation experiment at 277 K (Fig. 4a, b)[64,65], which generated 44, 48, and 37 datasets for eucalyptus, poplar, and spruce, respectively. This experiment used a $^1$H–$T_2$ relaxation filter of 120 μs × 2 to suppress the polysaccharide signals to <5% and retain >85% of water magnetization. Water-polarization was transferred to spatially proximal polymers using a 4 ms $^1$H mixing period, followed by a 1 ms CP for $^{13}$C detection. A 100 ms DARR mixing was used for both water-edited and control spectra. 1D buildup curves were measured at 277 K using a $^1$H–$T_2$ filter of 120 μs × 2 and a $^1$H mixing of 0–81 ms for spruce, 0–121 ms for eucalyptus, and 0–169 ms for poplar.

To probe dynamics, we measured $^{13}$C spin-lattice relaxation ($T_1$) and $^1$H rotating frame spin-lattice relaxation ($T_{1\rho}$). $^{13}$C–$T_1$ was measured using Torchia-CP[66] and standard $^{13}$C-DP inversion recovery schemes. Torchia-CP preferentially detected rigid molecules, with a z-filter of 0–12 s. A 30 s recycle delay was used in the DP inversion recovery experiment for quantitatively detecting all molecules, with a z-period of 0–35 s. $^1$H–$T_{1\rho}$ was measured using an effective $^1$H spin-lock field of 62.5 kHz with a duration of 0–19 ms. Relaxation data were fit using single exponential functions (Supplementary Tables 10 and 11). Also, the dipolar-chemical shift (DIPSHIFT) experiment[67] was conducted under 7.5 kHz MAS, to report dipolar order parameters of biopolymers in three wood samples. Frequency-Switched Lee Goldburg (FSLG) $^1$H homonuclear decoupling sequence was utilized, the scaling factor was verified to be 0.577.

**Solution-NMR sample preparation and experiments**. To validate the lignin assignment obtained using the solid-state method, we conducted solution-NMR analysis, which has well-documented chemical shifts of lignin polymers. A summary of the lignin chemical shifts is included in Supplementary Table 3. Wood samples were ball-milled for 1 h with a motor running at 1750 rpm. About 30 mg of each powdered wood sample was dissolved in 10 mL DMSO-$d_6$ with constant stirring using a magnetic stirrer for 2 h at 60 °C for solution-NMR experiments. To identify linkages in lignin, we conducted 2D $^1$H–$^{13}$C HSQC spectra conducted on a Bruker Avance III 500 MHz $^1$H Larmor frequency and equipped with a 5 mm $z$-gradient Prodigy TCI probe. Both the $^1$H and the $^{13}$C chemical shifts gave indications of the linkage types using the chemical shifts reported previously[68–73].

**MAS-DNP experiment**. About 30–35 mg of eucalyptus stems were impregnated into 50 μL DNP matrix solution containing 10 mM AMUPol[74] radical in $d_8$-gly-cerol/$D_2O$/$H_2O$ (60/30/10 Vol%). The material was softly ground for 10–15 min to allow radical distribution and packed into a 3.2 mm sapphire rotor. The NMR sensitivity was enhanced by 24-fold with microwave irradiation on a 600 MHz/ 395 GHz MAS-DNP spectrometer under 10 kHz MAS. DNP analysis has been applied to various carbohydrate and plant systems using different protocols[75–77]. Recently, we have demonstrated that the radical can effectively and homogeneously polarize all molecules in cell wall materials[78]. Lignin had a slightly shorter DNP buildup time (1.8 s) compared with polysaccharides (2.2 s). The recycle delay was set to 2.3 s to facilitate the selection of lignin against carbohydrates in the 2D experiments. A lignin-edited 2D $^{13}$C–$^{13}$C experiment was measured to detect the signals of lignin-bound carbohydrates, with a 0.5 s PDSD mixing to transfer lignin polarization to carbohydrates, followed by a 20-ms PDSD mixing for 2D $^{13}$C–$^{13}$C correlation[24]. For comparison, a 20 ms PDSD spectrum was measured as a control.

**DFT calculation**. DFT calculations were used to investigate the interactions between the S or G units in lignin and cellulose, or xylan with two- and three-fold conformations. Tetramers of β-1,4-linked glucose and xylose were built to represent cellulose and two-fold xylan, respectively. Hexamers of β-1,4-linked xylose were built to represent three-fold xylan, based on a crystal structure of three-fold xylopentaose (PDB ID: 1GNY)[79]. Six models were constructed: G unit-cellulose, G unit-xylan (two-fold), G unit-xylan (three-fold), S unit-cellulose, S unit-xylan (two-fold), and S unit-xylan (three-fold). Previously, it has been shown that for lignin–cellulose and lignin–xylan interactions, the stacked configurations tend to have the largest interaction energies[40]. Therefore, the initial starting configurations of the six models were constructed by positioning the G/S units parallel to the surface of the glucose/xylose residues, ~4 Å away. The geometry optimizations were conducted using the DFT method, M05-2X/6-31 + G*[80] with the integral equation formalism for the polarizable continuum model (IEFPCM)[81] solvation model in Gaussian 09[82]. Although other methods may be available, this exchange-correlation functional with dispersion corrections performs reasonably well for carbohydrate and carbohydrate-aromatic interactions[11,83,84].

**Reporting summary**. Further information on research design is available in the Nature Research Reporting Summary linked to this article.

## Data availability

All NMR spectra and biochemical data that support the findings of this study are provided in the article, Supplementary Information, Supplementary Data, and Source Data files. The processed topspin NMR datasets of plant stems are available upon requests from the corresponding author due to the large size of the files and lack of an appropriate repository for biopolymer/biomaterial NMR. The crystal structure of xylopentaose used in this study was previously published and is available as PDB entry 1GNY [https://doi.org/10.2210/pdb1GNY/pdb]. Source data are provided with this paper.

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

## Acknowledgements

The solid-state and solution-NMR analyses of wood cell walls were conducted by A.K., X.K., W.Z. and T.W., who were supported by the U.S. Department of Energy (grant no. DE-SC0021210). DFT calculation and compositional analysis conducted by H.Y. and F.D. were supported as part of the Center for Lignocellulose Structure and Formation, an Energy Frontier Research Center funded by the US Department of Energy, Office of Science, Basic Energy Sciences under award no. DE-SC0001090. The National High Magnetic Field Laboratory is supported by National Science Foundation through NSF/DMR-1644779 and the State of Florida. The MAS-DNP system at NHMFL is funded in part by NIH S10 OD018519 and NSF CHE-1229170. We thank Dr. Daniel Cosgrove for helpful discussion.

## Author contributions

A.K., X.K., F.M.V., F.D., W.Z. and T.W. conducted the NMR and DNP experiments. A.K., F.D., X.K. and W.Z. analyzed the experimental data. H.Y. performed DFT calculation. A.K. optimized the DNP samples. T.W. designed the project. All authors contributed to manuscript writing.

## Competing interests

The authors declare no competing interests.
