## [Peer Review File · Nature Communications]

REVIEWER COMMENTS

Reviewer #1 (Remarks to the Author):

The major claims in the paper are that (1) although three-fold xylan is favored for binding lignin, in wood, other xylan conformers can also coexist with lignin, (2) lignin and cellulose are colocalized in woody biomass, and (3) polymer entanglements and inter-penetration govern lignin-carbohydrate interactions in wood

The application of DNP NMR and the data interpretation of ^{13}C - ^{13}C RFDR, ^{13}C J-INADEQUATE, and ^{13}C - ^{13}C correlation PDSO experiment is quite complex and cutting edge and if believed lead to novel claims that have board interest to the community

The work done is meticulous; however, the chemical assignments in highly overlapping systems like cell wall are difficult often requiring reliance on well-established chemical shifts from highly regarded citations, chemical shifts from DFT modeling, as well as use of filtered NMR and NMR of isolated components. The present supporting information was insufficient to convince me the assignments in supplementary Table 1 and 2 were correct. The significant presence of extractive, proteins, and lipids (supplementary figure 3) make the chemical shift assignment and data interpretation of ^{13}C - ^{13}C RFDR, ^{13}C J-INADEQUATE, and ^{13}C - ^{13}C correlation PDSO experiment even more difficult to accept.

Some of the conclusions were difficult to understand due errors in word choice and writing but if I understood the authors correctly some conclusion are counter to our current understanding the secondary cell wall. This does not mean the authors are wrong but does mean their supporting effort around chemical shift assignment has to be unassailable and their interpretation need to be accompanied with reasonable plant cell wall biochemical and biosynthetic arguments

Minor notes:

Pg 2: Introduce the acronym before using "NMR"

Pg 2, Line 19 "Plant cell walls constitute the majority of lignocellulosic biomass and serve as an inexhaustible resource of biomaterials and biofuel" - Many may disagree that biomass is inexhaustible, it is abundant but not inexhaustible.

Pg 3, Line 36 "Lignification mechanically strengthens secondary walls but also renders the feedstock resistant to enzymatic hydrolysis during the conversion to liquid transportation fuel" This is not explicitly true. There are a number of factors that make biomass recalcitrant and lignification is only related to some sub-set of those factor.

Pg 3, Line 40 "However, these efforts have to take place empirically due to our limited understanding of cell wall architecture" Again this is not explicitly true. I think there is a lot to be learned about cell wall architecture that could further enable rationale design of genetic modifications or processing but to say that effort until now have been largely empirically driven would be incorrect.

Pg 3, Line 43 "The secondary cell wall is assembled by carbohydrate and aromatic constituents, whose structural complexity is beyond anything artificially producible" I would argue this is an imprecise statement. Yes the cell wall is complex and designed by nature with specific structure-function relationships difficult to replicate, but there are a number of artificial structures that rival the complexity of the cell wall.

Pg 3, Line 44 "Each elementary cellulose microfibril contains eighteen 1,4- β -glucan chains, which are held together by a hydrogen-bonding network and likely arranged in a six-layered organization, with 2, 3, 4, 4, 3, and 2 chains in each layer" It is fine to assume a fibril configuration for your own analysis but as far as I know the exact structure of cellulose microfibril is unresolved with several probably configurations 24-chain diamond-shaped model, 2.9 x 2.7 nm; 24-chain rectangular model, 3.3 x 2.7 nm; 18-chain rectangular model, 2.5 x 2.7 nm.

Pg 3: Hemicellulose in cell wall can be xylan, glucuronoxylan, arabinoxylan, or glucomannan.

Pg 3: What is meant by "sophisticated" covalent linkers. Although sophisticated can be used to imply complexity, it is typically used to describe someone with a great deal of worldly experience and knowledge of fashion and culture. Moreover, I would argue is someone was not familiar with what hemicellulose or lignin was they still are not. I understand space limitations, so advocate for more accurate and precise language with the express purpose of defining cellulose, hemicellulose and lignin.

Pg 3, Line 55: "This conjecture is outlined by diffraction data that focus on the spatial arrangement of cellulose microfibrils, imaging results that map out the cell wall meshes and the microscopic distribution of lignin, and solution NMR spectra that identify the lignin-carbohydrate bonding". It is not clear what the opening "This conjecture" is referring to and I believe supported works better that outline. This sentence, in general, could be written better.

Pg 3: What is a "packing interface" vs "interface"?

Pg 3: The interface between lignin and polysaccharides is difficult probe due to the length scale but also because it is not a prevalent bulk phenomena, requiring deconvolution from rest of the cell wall, they exist in the solid-state, and separation methods to isolate easily alter lignin and polysaccharides interaction.

Pg 4: What is a "surface contact" interaction? Maybe hydrogen bonding or hydrophobic-hydrophobic interactions.

Pg 4, Line 68: What is meant by "three prime doubts about the generality of"

Pg 4, Line 77: What is meant by "mechanical joints"

Pg 4, Line 78: It not clear what is meant by "Wood" molecules

Pg 4, Line 80: There are a lot of models and data that also show that the cell wall architecture is intimately mixed vs existing as a separated domain

What species/variants of poplar, eucalyptus, and spruce are being used? How old are the plants? Where is the stem sampled from on the plant? Were stems de-barked? How was grinding done? Were extractives removed? If not, small molecule extractive especially those that are protein, sugars, and aromatics that could overlap with and obscure cell wall chemical shift could be problematic.

Freeze drying and then rehydrating could be problematic. Complete drying can cause hornification and cell wall collapse that is irreversible. The hydration and dynamics results maybe altered due to this.

Not clear how chemical shift assignments were done. Was literature used or did you have a protocol to establish chemical shifts in Supplementary Table 1 and 2.

It's hard to believe hardwood lignin and polysaccharides is not well-mixed on length scale detectable by spin diffusion. Maybe the spin diffusion coefficients are very different and hardwoods need 0.2 or 0.3s to match the spruce at 0.1s.

Pg 11, Line 257 "electrostatic interactions are essential to the existence of lignin-xylan complex." Attractive electrostatic interaction generally involve positive-negative charges, that not what you are referring too.

Pg 13, Line 321 "these two macromolecules are spaced by xylan in Arabidopsis and grasses but become colocalized in woody biomass" I am not sure what this means, are you saying unlike grasses lignin-cellulose contact exist without the presence of hemicellulose?

Further detailing of Figure 5 in a conclusion type paragraph would be helpful

Reviewer #2 (Remarks to the Author):

It is an important work investigating supramolecular structures of wood lignocellulose using sophisticated solid-state NMR approaches, providing important basis for understanding the molecular architecture of plant cell walls and instructive information for utilizing them as sustainable materials. In particular, the authors successfully detected, primarily aided by DNP, long-range interactions defining the associations between lignin and carbohydrate components in intact cell walls, and then, along with polymer dynamics data based on ^1H transfer polarization and T1 relaxation experiments, dissect their interesting variations among lignocellulose samples from different wood plant species, i.e., softwood vs hardwood, as well as grasses (using literature data). The quality of each experiment and data presentation is superb (although some concerns as

described below), and the manuscript is overall well-written. While I truly value this work, I have several concerns.

Major comments:

I really appreciate that the authors determine lignocellulose composition in intact cell walls using sophisticated solid-state NMR methods. Nevertheless, I insist the author should report statistic lignocellulose composition data determined by conventional methods using wet-chemistry and/or semi-quantitative solution-state HSQC NMR approaches (with ball-milled cell wall samples) to test whether or not the current data based on solid-state NMR methods (fig 1g for polysaccharides and fig. 2d and 2e for lignin) are consistent with those determined by the conventional methods, and if not, discuss what makes the inconsistency.

In particular, there seems to be a considerable discrepancy in the lignin chemistry reported here (fig. 2e by solid-state and 2f by solution-state NMRs) given what we usually expect for lignins in the three very typical hardwood and softwood species (see, for example, Scheme 5 in Rinaldi et al. *Angew. Chem. Int. Ed.* 55, 8164, 2016).

The most prominent example is the abundance of β -ether (A) reported in fig. 2e. While this linkage has been well-recognized as the most predominant linkage type in natural lignins, accounting for more than 50% (very at least) regardless of the plant source, the authors reported very low β -ether abundances (19% and 14%) for eucalyptus and spruce lignins, albeit rather normally (66%) for poplar lignin (fig. 2e); the authors mention that these data are consistent with their solution-state HSQC NMR data (fig. 2f) but the quality of the HSQC spectra look not so good... Also, usually, dibenzodioxocin (D) is more abundant in softwood than in hardwood (because it arises from G lignin polymerization but not from S lignin polymerization), but the authors got totally opposite results, and unusually high numbers in hardwood eucalyptus and poplar. Please verify these data and, if they are correct, please explain what can cause these. I am not sure if these are because of the use of CP-based experiments. Can the authors detect (and also quantify) these lignin signals in DP-based experiments as well?

Another intriguing but weird thing I noticed was the detection of benzodioxane (V) in both eucalyptus and spruce stem lignins (fig. 2e). Stilbene-derived benzodioxanes were recently identified in lignin from the "bark" of Norway spruce (Rencoret et al., *Plant Physiol.* 180, 1310-1321, 2019). I don't think there is any report noting their existence in "de-barked stems" of spruce like what the

authors reported here (18%). It is also very unusual that natural (non-transgenic) eucalyptus lignin contains benzodioxane that much (13%) without incorporating stilbene or any other unusual lignin monomers. I suspect the peak assignments are wrong. Unless the authors can provide more convincing NMR data to prove their existence, I recommend omitting this particular linkage type from the current analysis.

Minor suggestions:

Line 252: "two methyl ether groups" -> "two methoxyl groups"

Line 254: "methyl ether end" -> "methoxyl group" or "single methoxyl group"

Line 255: "methyl groups" -> "methoxyl groups"

Line 256: "the methyl-rich S-residues" -> "the methoxyl-rich S-residues"

Line 359 (Methods for plant material preparation): Here or in Introduction, please provide complete scientific names of the three wood species to clarify specifically what eucalyptus, poplar and spruce species were used.

Reviewer #3 (Remarks to the Author):

In this work by Kirui et al. the authors use now standard solid-state NMR methods to explore the interaction between lignin and various polysaccharides in higher plant cell walls. The work provides major findings on the structure, dynamics, hydration and mixing of cellulose, xylans and lignin for three species of trees.

The methodology is adequate, experiments well designed and the article reads well despite the abundance and complexity of the information presented. The figures are high quality and clear.

I believe these results to be of great interest to the readership of Nature Communications and recommend it for publication after very minor revisions.

Some suggestions:

I find figure 5 a little hard to read. Perhaps it could be improved by adding a color identification in the figure rather than in the caption (ex: lignin: yellow rectangle). The authors could even consider keeping a consistent color coding for the main components (lignin, cellulose, xylans...) throughout the article. I am not sure that the topview and sideview are necessary and find the sideview clearer.

The different plants are harvested at different growth times (16 to 27 weeks) and two are described as seeds while one is a stem. Could the authors comment on the growth stage of the different samples and the potential molecular changes that might occur during plant growth?

Minor points:

Line 113: "The remarkable resolution is signified by the narrow ^{13}C linewidths of 0.5-0.9 ppm... " I don't think "signified" is the right verb.

Line 260: "Biopolymers sophisticated dynamics and water-association in their native environments." Why this sentence?

Lines 291-293: "When all molecules were considered, the ^{13}C -T1, as well as the characteristic nanosecond motions, decreased in the order of cellulose, lignin, xylan, and mannan, if any." This sentence might confuse the reader, a decrease in T1 (increase in relaxation efficiency) indicates an increase in nanosecond motions.

Summary: We would like to thank the reviewers for providing helpful advice regarding the aspects of cell wall biochemistry, NMR spectroscopy, as well as lignin biochemistry. Following the instructions, we have substantially improved the manuscript by **the addition of a new maintext Figure, five Supplementary Figures, and five Supplementary Tables**, in addition to the replotting of the existing figures and the rephrasing of the writing. Notable changes include the addition of more details regarding resonance assignment (e.g., detailed comparison with literature in Supplementary Tables 5 and 19) and samples preparation, inclusion of new dataset collected on never-dried samples (Fig. 5 and Supplementary Fig. 16), editing and shortening the lignin section, inclusion of more relevant literature, and description of the current technical limitation. We hope the revision helps to increase the clarity and accessibility of the data and findings.

Responses to Reviewers

Reviewer #1 (Remarks to the Author):

The major claims in the paper are that (1) although three-fold xylan is favored for binding lignin, in wood, other xylan conformers can also coexist with lignin, (2) lignin and cellulose are colocalized in woody biomass, and (3) polymer entanglements and inter-penetration govern lignin-carbohydrate interactions in wood

The application of DNP NMR and the data interpretation of ¹³C-¹³C RFDR, ¹³C J-INADEQUATE, and ¹³C-¹³C correlation PDSM experiment is quite complex and cutting edge and if believed lead to novel claims that have board interest to the community

We appreciate the encouraging comments about the method and the significance of the results.

The work done is meticulous; however, the chemical assignments in highly overlapping systems like cell wall are difficult often requiring reliance on well-established chemical shifts from highly regarded citations, chemical shifts from DFT modeling, as well as use of filtered NMR and NMR of isolated components. The present supporting information was insufficient to convince me the assignments in supplementary Table 1 and 2 were correct. The significant presence of extractive, proteins, and lipids (supplementary figure 3) make the chemical shift assignment and data interpretation of ¹³C-¹³C RFDR, ¹³C J-INADEQUATE, and ¹³C-¹³C correlation PDSM experiment even more difficult to accept.

We would like to thank the reviewer for the very helpful advices regarding the resonance assignment. Several steps have been undertaken to improve the technical clarity. First, we measured new 1D spectra of spruce and eucalyptus (**Fig. 5** and **Supplementary Fig. 16**), showing that proteins and lipids have well-resolved NMR spectral regions and also have minor contribution to the rigid phase (the same for small sugars and extractives) that we are detecting. Secondly, we added a flow chart (**Supplementary Figure 17**) as an introduction to our assignment protocol. All the chemical shifts are compared with literature reported values in **Supplementary Table 19**. While the assignment of polysaccharides is really straightforward and were based on the existing literature deposited in database, the lignin assignment is more challenging and relies on the extensive solution NMR studies. Therefore, we compiled more references from solution NMR and added new **Supplementary Table 4** and **5**. More detailed responses are also provided below for the specific question regarding the signals of proteins, lipids, and extractives. Also, the key technical parameters of the experiments are now added as **Supplementary Tables 16-18**.

Addition of the supporting data should improve the clarity regarding resonance assignment.

Some of the conclusions were difficult to understand due errors in word choice and writing but if I understood the authors correctly some conclusion are counter to our current understanding the secondary cell wall. This does not mean the authors are wrong but does mean their supporting effort around chemical shift assignment has to be unassailable and their interpretation need to be accompanied with reasonable plant cell wall biochemical and biosynthetic arguments

Thanks for providing specific and helpful comments. We have addressed all of them to describe the concepts and findings more accurately, which improves the accuracy of the writing and better connect with the biochemical concepts and current understanding. We have also included more discussion of recent ssNMR studies of secondary cell walls (e.g., by the research groups of Dr. Mortimer and Dr. Dupree) to better align this study with the ongoing efforts in understanding secondary cell wall structure. Fifteen new references have been added to better acknowledge the research field. Also, we have now provided a substantially expanded dataset to provide support to the major findings, e.g. by excluding the disruption from different spin diffusion coefficients and assessing the effect of rehydration. The changes are detailed below in response to the specific queries.

Minor notes:

Pg 2: Introduce the acronym before using “NMR”

Thanks. We have changed it to “nuclear magnetic resonance (NMR)”

Pg 2, Line 19 “Plant cell walls constitute the majority of lignocellulosic biomass and serve as an inexhaustible resource of biomaterials and biofuel” - Many may disagree that biomass is inexhaustible, it is abundant but not inexhaustible.

We now use ‘renewable’ to describe the plant biomass.

Pg 3, Line 36 “Lignification mechanically strengthens secondary walls but also renders the feedstock resistant to enzymatic hydrolysis during the conversion to liquid transportation fuel” This is not explicitly true. There are a number of factors that make biomass recalcitrant and lignification is only related to some sub-set of those factor.

Thanks for the helpful comment. We have now rewritten this sentence to reflect that the presence of lignin and its interaction with polysaccharides are among the contributors of biomass recalcitrance: “Lignification mechanically strengthens secondary walls, however, the presence of these intractable polyphenols and their association with carbohydrate components contributes to the biomass recalcitrance that renders the feedstock resistant to enzymatic hydrolysis during its conversion to liquid transportation fuel.”

Pg 3, Line 40 “However, these efforts have to take place empirically due to our limited understanding of cell wall architecture” Again this is not explicitly true. I think there is a lot to be learned about cell wall architecture that could further enable rationale design of genetic modifications or processing but to say that effort until now have been largely empirically driven would be incorrect.

We now better described it as “These efforts have not yet reached the full potential due to our limited understanding of cell wall architecture.”

Pg 3, Line 43 “The secondary cell wall is assembled by carbohydrate and aromatic constituents, whose structural complexity is beyond anything artificially producible” I would argue this is an imprecise statement. Yes the cell wall is complex and designed by nature with specific structure-function relationships difficult to replicate, but there are a number of artificial structures that rival the complexity of the cell wall.

We have now provided a more precise statement: “The secondary cell wall is assembled by carbohydrate and aromatic constituents, with remarkable complexity and variability.”

Pg 3, Line 44 “Each elementary cellulose microfibril contains eighteen 1,4- β -glucan chains, which are held together by a hydrogen-bonding network and likely arranged in a six-layered organization, with 2, 3, 4, 4, 3, and 2 chains in each layer” It is fine to assume a fibril configuration for your own analysis but as far as I know the exact structure of cellulose microfibril is unresolved with several probably configurations 24-chain diamond-shaped model, 2.9 x 2.7 nm; 24-chain rectangular model, 3.3 x 2.7 nm; 18-chain rectangular model, 2.5 x 2.7 nm.

We fully agree with the reviewer. We have improved the clarity by rewriting this sentence: “Each elementary cellulose microfibril contains eighteen 1,4- β -glucan chains, which are held together by a hydrogen-bonding network. The exact organization of these glucan chains is unresolved, but recent density functional theory (DFT) calculations suggest a six-layered organization, likely with 2, 3, 4, 4, 3, and 2 chains in each layer (Fig. 1a).”

Pg 3: Hemicellulose in cell wall can be xylan, glucuronoxylan, arabinoxylan, or glucomannan.

We have rewritten this sentence: “Hemicelluloses, such as xylan, glucuronoxylan, arabinoxylan, and glucomannan, are highly variable in their monosaccharide composition and linkage pattern.”

Pg 3: What is meant by “sophisticated” covalent linkers. Although sophisticated can be used to imply complexity, it is typically used to describe someone with a great deal of worldly experience and knowledge of fashion and culture. Moreover, I would argue is someone was not familiar with what hemicellulose or lignin was they still are not. I understand space limitations, so advocate for more accurate and precise language with the express purpose of defining cellulose, hemicellulose and lignin.

Thanks for the helpful advice, we have expanded the introduction of lignin: “Lignin contains guaiacyl (G), syringyl (S), and p-hydroxyphenyl (H) phenolic residues, which are interconnected by different types of covalent linkers such as β -O-4 ether-O-aryl, β - β ' resinol, and β -5' phenylcoumaran.”

The introduction of cellulose and hemicellulose has also been improved following the instruction, as detailed in the responses to the previous comments.

Pg 3, Line 55: “This conjecture is outlined by diffraction data that focus on the spatial arrangement of cellulose microfibrils, imaging results that map out the cell wall meshes and the microscopic distribution of lignin, and solution NMR spectra that identify the lignin-carbohydrate bonding”. It is not clear what the opening “This conjecture” is referring to and I believe supported works better that outline. This sentence, in general, could be written better.

We have rewritten this sentence for clarity: “Our understanding of cell wall organization is supported by many studies that employed diffraction methods to reveal the spatial arrangement of cellulose

microfibrils, imaging techniques to map out cell wall meshes and the microscopic distribution of lignin, and solution NMR spectroscopy to identify lignin-carbohydrate linkages”

Pg 3: What is a “packing interface” vs “interface”?

To avoid confusion, we removed “packing” and only mention the polymer interface between lignin and polysaccharides.

Pg 3: The interface between lignin and polysaccharides is difficult probe due to the length scale but also because it is not a prevalent bulk phenomena, requiring deconvolution from rest of the cell wall, they exist in the solid-state, and separation methods to isolate easily alter lignin and polysaccharides interaction.

This is a very insightful point! We have now included it: “However, the interface between lignin and polysaccharides, the focus of this study, is not yet well understood. This is partially due to the hardly accessible length scale (angstrom to nanometer) and the requirement of both chemical and atomic resolutions. In addition, only a small number of molecules reside on this lignin-carbohydrate interface, which needs to be deconvoluted from the bulk of the cell wall. As both lignin and polysaccharide exist in the solid state, conventional separation methods often perturb their structures and interactions, making it difficult to investigate this polymer interface.”

Pg 4: What is a “surface contact” interaction? Maybe hydrogen bonding or hydrophobic-hydrophobic interactions.

It is to distinguish from polymer entanglement and penetration. We have now rephrased it for clarity: “Lignin tends to form hydrophobic and disordered nanodomains, the surface of which binds the xylan in a three-fold helical screw conformation.”

Pg 4, Line 68: What is meant by “three prime doubts about the generality of”

We reformulated this sentence to “To generalize these structural principles, we need to examine other plant systems to evaluate three critical aspects: (i) the conformational bias of hemicellulose’s function, (ii) the absence of cellulose-lignin contact, and (iii) the self-aggregating nature of aromatic polymers.”

Pg 4, Line 77: What is meant by “mechanical joints”

We have better described it as “the junctions of cellulose surface and flat-ribbon xylan.”

Pg 4, Line 78: It not clear what is meant by “Wood” molecules

We now give a clearer description: “In these woody plants, molecules experience a highly homogeneous mixing....”

Pg 4, Line 80: There are a lot of models and data that also show that the cell wall architecture is intimately mixed vs existing as a separated domain.

Thanks for pointing it out. We have removed the improper statement of “revising the prevailing paradigms based on domain separation” on **Page 4**.

What species/variants of poplar, eucalyptus, and spruce are being used?

For poplar we used a stem cutting as starting material, the mother plant is identified as *Populus x canadensis*. Seeds from Eucalyptus (*Eucalyptus grandis*) and Spruce (*Picea abies*) were obtained from tree-seed. It is now clarified on **Page 4**.

How old are the plants?

Poplar wood was 27 weeks old, eucalyptus was 16 weeks, and spruce was 16 weeks. Poplar was obtained from stem cutting while eucalyptus and spruces were obtained from seeds. The information is provided in the Methods section (Plant material) on **Page 18**.

Where is the stem sampled from on the plant?

Poplar plants were saplings of close to 1 m height at harvest. Immediately after removing the plants from the growth chamber, plant shoots were dissected into leaves and stems. The stem-sections were split and cut and debarked after freeze-drying. We have now clarified it in in the Methods section on **Page 18**.

Were stems de-barked?

Yes, and the description of the de-barking process is now provided on **Page 18**.

How was grinding done?

We have now added more details in the Methods section (**Page 18**). The material was briefly hand-grinded using a pestle and a mortar, resulting in small pieces on the range of a few mm across. The homogeneity helps to avoid potential issues during magic-angle spinning (to minimize the chance of rotor crash).

Were extractives removed? If not, small molecule extractive especially those that are protein, sugars, and aromatics that could overlap with and obscure cell wall chemical shift could be problematic.

We did not remove extractives. With the current spectral resolution of biomolecular NMR, we can track the carbon connectivity in most types of molecules. Therefore, the extractives have relatively minor effect and intact tissue materials are widely used for investigating secondary cell walls. Examples include the pioneering studies by Mortimer, Dupree, and Hong (e.g., Gao et al. *Nat Commun.* 11, 6081, 2020; Terrett et al. *Nat Commun* 10, 4978, 2019, Duan et al. *ACS Omega*, 6, 15460-471, 2021). The proteins, small sugars, and lipid polymers are mobile components in the cell wall, thus not giving negligible signal here using cross polarization methods that select rigid components. The very low intensities of proteins and lipid polymers in CP-based spectra can be directly visualized in their low intensities in corresponding 1D ^{13}C spectra (e.g., in the new **Supplementary Fig. 3a-c**). The chemical shifts of amino acids and lipids are well documents, and they also either have distinct chemical shifts or correlations (e.g., we can resolve all amino acids by tracking their carbon connectivity of $\text{C}\alpha\text{-C}\beta\text{-C}\gamma\text{...}$) from polysaccharides and lignin or is spatially separated (and thus will not obscure the findings of long-range correlations). Small sugars have unique ^{13}C chemical shifts, which are typically smaller than those of polysaccharides. This is also the beauty of high-resolution multidimensional NMR spectra where different biomolecules have their own resolvable spectral fingerprint.

Freeze drying and then rehydrating could be problematic. Complete drying can cause hornification and cell wall collapse that is irreversible. The hydration and dynamics results maybe altered due to this.

We have now added a **new section** “Effects of rehydration on water-retention and polymer dynamics” on **Pages 14-16** and two new Figures (**Figure 5** and **Supplementary Figure 16**), to show that, on the molecular level examined by ssNMR, the vast majority of polymer dynamics, structure, and hydration are largely retained after freeze-drying and rehydration of these woody species. The sample and data are freshly acquired; we did it with care by first measuring a never-dried sample, and then use the same sample for dehydration and rehydration, followed by another measurement.

In the same section, we have also highlighted the molecular changes observed in the grass *Sorghum bicolor* reported in the landmark study by Dr. Mortimer’s group, where dynamical changes were spotted in the arabinose sidechains of xylan and its 3-fold backbone. Also, to bring to the reviewers’ attention, a beautiful study (Cresswell et al. Biomacromolecules, in press) was published during the revision of this manuscript, detailing the changes of polymer packing and dynamics by complete oven-drying and rehydration in a softwood *Pinus radiata*.

Therefore, we thank the reviewer for this insightful point. And we hope our expansion regarding the water-retention can bring the manuscript closer to the frontier of science.

Not clear how chemical shift assignments were done. Was literature used or did you have a protocol to establish chemical shifts in Supplementary Table 1 and 2.

It’s hard to believe hardwood lignin and polysaccharides is not well-mixed on length scale detectable by spin diffusion. Maybe the spin diffusion coefficients are very different and hardwoods need 0.2 or 0.3s to match the spruce at 0.1s.

We have now added a new **Supplementary Figure 17** to show the flowchart of the assignment. Assignments require going through the complete carbon connectivity of each sugar unit and structural motif. It is quite straightforward to identify the assignments of carbohydrates using our knowledge e of these molecules and based on the chemical shifts deposited in the CCMRD database. The lignin assignment is more challenging, and we now provided new tables (**Supplementary Tables 4, 5 and 19**) to rely on the extensive solution NMR literature to guide the assignment.

We performed additional experiments (new **Supplementary Figure 7 and 8**) confirming that the observed equilibrium in spruce is not due to spin diffusion coefficient. The dipolar order parameters are comparable in 3 wood samples, for cellulose, hemicellulose and lignin. The spin diffusion rate of lignin and carbohydrates are also comparable in all three woods as confirmed by the similar spectral patterns observed for cellulose, xylan, and lignin intramolecular cross peaks. More importantly, it is clear that spruce showed relatively smaller order parameter and polarization transfer for some xyans, which, by expectation, should hinder the polarization transfer and give slower equilibrium, which goes against the observed rapid equilibrium. This confirmed and further strengthened the evidence that spruce molecules are well mixed on the nanoscale.

We have now clarified this point: “This is not caused by variations of spin diffusion coefficients or polymer dynamics in hardwood and softwood as validated experimentally (Supplementary Figs. 7 and 8). Therefore, the rapid equilibrium observed in spruce indicates that lignin and polysaccharides are well-mixed on the nanoscale in spruce but stay apart in hardwoods.”

Pg 11, Line 257 “electrostatic interactions are essential to the existence of lignin-xylan complex.” Attractive electrostatic interaction generally involve positive-negative charges, that not what you are referring too.

We apologize for the confusion. Electrostatic interaction, in a narrower way of definition, is about forces (both attractive and repulsive) between two electrically charged particles, ruled by Coulomb's law. However, in this manuscript, we are using its general definition, also including dipole-dipole interaction (between two dipolar molecules through space), London dispersion forces (between atoms and molecules that are normally electrically symmetric), hydrogen bonding (stronger than dipole-dipole interaction but comparatively weaker than covalent and ionic bonds), etc. And dipolar couplings are broadly utilized and detected in ssNMR. Hence, we are not assuming which kind of electrostatic interactions are exist in lignin-xylan complex but stating that is a kind of non-covalent interactions between two dipolar molecules through space.

To avoid confusion, we have modified this sentence to “Therefore, non-covalent interactions between these polar groups are essential to the existence of lignin-xylan complex.”

Pg 13, Line 321 “these two macromolecules are spaced by xylan in Arabidopsis and grasses but become colocalized in woody biomass” I am not sure what this means, are you saying unlike grasses lignin-cellulose contact exist without the presence of hemicellulose?

In grass and Arabidopsis, lignin and cellulose are separated, mainly held together in the wall by xylan. The situation has changed for woody biomass, in which direct lignin-cellulose contacts have become possible. We have now clarified it on **Page 16**.

Reviewer #2 (Remarks to the Author):

It is an important work investigating supramolecular structures of wood lignocellulose using sophisticated solid-state NMR approaches, providing important basis for understanding the molecular architecture of plant cell walls and instructive information for utilizing them as sustainable materials. In particular, the authors successfully detected, primarily aided by DNP, long-range interactions defining the associations between lignin and carbohydrate components in intact cell walls, and then, along with polymer dynamics data based on ^1H transfer polarization and T_1 relaxation experiments, dissect their interesting variations among lignocellulose samples from different wood plant species, i.e., softwood vs hardwood, as well as grasses (using literature data). The quality of each experiment and data presentation is superb (although some concerns as described below), and the manuscript is overall well-written. While I truly value this work, I have several concerns.

We would like to thank the reviewer for the encouraging comments. We also appreciate the very helpful suggestions on how to improve the sections related to lignin characterization, which helped us to reduce the ambiguity of the manuscript.

Major comments:

I really appreciate that the authors determine lignocellulose composition in intact cell walls using sophisticated solid-state NMR methods. Nevertheless, I insist the author should report statistic lignocellulose composition data determined by conventional methods using wet-chemistry and/or

semi-quantitative solution-state HSQC NMR approaches (with ball-milled cell wall samples) to test whether or not the current data based on solid-state NMR methods (fig 1g for polysaccharides and fig. 2d and 2e for lignin) are consistent with those determined by the conventional methods, and if not, discuss what makes the inconsistency.

Thanks a lot! We fully agree with the reviewer that solid-state NMR have rarely been used for quantifying lignin composition, in particular for the complex linkage pattern. Therefore, we made multiple major changes below: First, we have added a new panel in **Fig. 2d** to directly compare the solid-state NMR estimation of lignin composition with literature reported values from solution NMR (also in **Supplementary Table 19** and **5**). Second, we realize that despite the identification of a few key linkages, solid-state NMR is still very immature for characterizing lignin linkage, which will also be of great interest for more exploratory studies. Therefore, we removed the quantification of linkage and have pointed out this limitation on **Page 8** with the purpose of inspiring further method development, likely in collaboration with experts in lignin wet chemistry: “These results demonstrated the feasibility of using solid-state NMR to characterize lignin structure and linkage, but the technical capability still requires further development.” This also helped making this manuscript more concentrated on the spatial organization and interface of lignin-carbohydrate.

In particular, there seems to be a considerable discrepancy in the lignin chemistry reported here (fig. 2e by solid-state and 2f by solution-state NMRs) given what we usually expect for lignins in the three very typical hardwood and softwood species (see, for example, Scheme 5 in Rinaldi et al. *Angew. Chem. Int. Ed.* 55, 8164, 2016).

First, we have checked and revised our assignment and clarified the process in Supplementary Methods. We have substantially improved this part by adding three supplementary tables and a new figure (**Supplementary Figure 3** and **Tables 4, 5, and 19**) using the literature-reported and our freshly measured solution NMR HSQC spectra to guide the verification of solid-state NMR results. Also, as quantification is not a major goal here, we have removed the original panel of **Fig. 2** that was related to the composition of lignin linkage. The lignin section is substantially reduced to make the manuscript more focused and reduce uncertainty. The suggested reference is also used for cross validation and is now added to the references.

In addition, we have added many new references to both the Maintext and Supplementary Information (most of which are related to lignin NMR literature), which helps to better acknowledge the solution-NMR studies of lignin and biomass.

The most prominent example is the abundance of β -ether (A) reported in fig. 2e. While this linkage has been well-recognized as the most predominant linkage type in natural lignins, accounting for more than 50% (very at least) regardless of the plant source, the authors reported very low β -ether abundances (19% and 14%) for eucalyptus and spruce lignins, albeit rather normally (66%) for poplar lignin (fig. 2e); the authors mention that these data are consistent with their solution-state HSQC NMR data (fig. 2f) but the quality of the HSQC spectra look not so good...

We have reprocessed the HSQC spectra with equal apodisation window functions and appropriate forward linear prediction in the indirect dimension, which has been detailed now in the Methods section. We have now provided a new **Supplementary Tables 5 and 19** to tabulate the literature-reported solution NMR chemical shifts for direct comparison with the solid-state NMR values. We have removed the estimation of lignin linkage composition and also revised the related assignment by only reporting the major types confirmed by comparison with literature results.

Also, usually, dibenzodioxocin (D) is more abundant in softwood than in hardwood (because it arises from G lignin polymerization but not from S lignin polymerization), but the authors got totally opposite results, and unusually high numbers in hardwood eucalyptus and poplar. Please verify these data and, if they are correct, please explain what can cause these. I am not sure if these are because of the use of CP-based experiments.

Thanks for catching the issue. We apologize that dibenzodioxocin was mistakenly reported, where it actually corresponded to traces of spirodienone. We have now double checked and revised the tables and maintext to avoid mismatching. Comparison with literature values is also provided in **Supplementary Tables 5 and 19**.

Can the authors detect (and also quantify) these lignin signals in DP-based experiments as well?

Following the advice, we have acquired a new set of DP-based spectra and provided it as **Supplementary Figure 3**. Most lignin signals are detectable in both CP- and DP-based experiments, demonstrating their distribution in both rigid and mobile phases.

Another intriguing but weird thing I noticed was the detection of benzodioxane (V) in both eucalyptus and spruce stem lignins (fig. 2e). Stilbene-derived benzodioxanes were recently identified in lignin from the “bark” of Norway spruce (Rencoret et al., Plant Physiol. 180, 1310-1321, 2019). I don’t think there is any report noting their existence in “de-barked stems” of spruce like what the authors reported here (18%). It is also very unusual that natural (non-transgenic) eucalyptus lignin contains benzodioxane that much (13%) without incorporating stilbene or any other unusual lignin monomers. I suspect the peak assignments are wrong. Unless the authors can provide more convincing NMR data to proof their existence, I recommend omitting this particular linkage type from the current analysis.

We thank the reviewer for the really helpful advice. We have now removed this linkage type from the current analysis and also substantially reduced the content related to lignin linkage. As the first step, we feel that pointing out that solid-state NMR can track some of the linkage might already be highly novel. More detailed analysis should be done later using well-controlled samples/systems, and with the assistance from experts in solution NMR analysis and lignin chemistry/biochemistry. However, this is for sure a highly important and promising direction.

Minor suggestions:

Line 252: “two methyl ether groups” -> “two methoxyl groups”

Line 254: “methyl ether end” -> “methoxyl group” or “single methoxyl group”

Line 255: “methyl groups” -> “methoxyl groups”

Line 256: “the methyl-rich S-residues” -> “the methoxyl-rich S-residues”

Thanks for the instruction. We have made all the corrections. In addition, we have now updated to “methoxyl” at each of its occurrence throughout the manuscript.

Line 359 (Methods for plant material preparation): Here or in Introduction, please provide complete scientific names of the three wood species to clarify specifically what eucalyptus, poplar and spruce species were used.

We have now included the scientific names in the Introduction (**Page 4**) and in the Methods section (**Page 18**). The wood samples used for this study was *Eucalyptus grandis*, *Populus x canadensis*, and

Picea abies. In addition, more technical details are now provided regarding the sample preparation in the Methods section for better clarity.

Reviewer #3 (Remarks to the Author):

In this work by Kirui et al. the authors use now standard solid-state NMR methods to explore the interaction between lignin and various polysaccharides in higher plant cell walls. The work provides major findings on the structure, dynamics, hydration and mixing of cellulose, xylans and lignin for three species of trees.

The methodology is adequate, experiments well designed and the article reads well despite the abundance and complexity of the information presented. The figures are high quality and clear.

I believe these results to be of great interest to the readership of Nature Communications and recommend it for publication after very minor revisions.

We really appreciate the encouraging comments. We have now added more evidence to support the major conclusions and improve the technical clarity. We have also remade the model figure (now **Fig. 6**) and rewritten the associated discussion to efficiently convey the structural findings. We hope the study could be of interest to our research community.

Some suggestions:

I find figure 5 a little hard to read. Perhaps it could be improved by adding a color identification in the figure rather than in the caption (ex: lignin: yellow rectangle). The authors could even consider keeping a consistent color coding for the main components (lignin, cellulose, xylans...) throughout the article. I am not sure that the topview and sideview are necessary and find the sideview clearer.

We have replotted the structural scheme (now **Fig. 6**). Following the suggestion, only the sideview is kept. Color coding is provided for the main components (lignin, cellulose, 2- and 3-fold xylan, and GGM). To make the figure less crowded, we have now provided the zoom-in structural views of these molecules as separate and numbered panels underneath the main schemes. We hope these changes help to improve the accessibility of the figure and the structural deliveries.

The different plants are harvested at different growth times (16 to 27 weeks) and two are described as seeds while one is a stem. Could the authors comment on the growth stage of the different samples and the potential molecular changes that might occur during plant growth?

Thanks for the insightful advice. First, we have now added more details about the plant growth and sample preparation in the Methods section (**Page 18**). Second, we have rewritten the last paragraph of the Discussion section (**Pages 18-19**) to mention that while the current study focuses on the physical and chemical principles regulating polymer packing in lignocellulose, more in-depth investigations are needed for understanding the structural variations presented by numerous plant species, mutants, cell types, and growth stages. Third, we have added a new section “Effects of Plant Age on Cell Walls” in the **Supplementary Methods** for a brief discussion of this aspect. We hope the ssNMR strategy and the fundamental structural principles presented in this study will prepare and inspire us and many other colleagues to address the challenging questions related to lignocellulose.

Minor points:

Line 113: “The remarkable resolution is signified by the narrow ^{13}C linewidths of 0.5-0.9 ppm... “ I don’t think “signified” is the right verb.

We have changed it to “The remarkable resolution is evidenced by the narrow ^{13}C linewidths.”

Line 260: “Biopolymers sophisticated dynamics and water-association in their native environments.”
Why this sentence?

We have corrected the error: “Biopolymers have sophisticated dynamics and variable water-association in their native environments”.

Lines 291-293: “When all molecules were considered, the ^{13}C -T₁, as well as the characteristic nanosecond motions, decreased in the order of cellulose, lignin, xylan, and mannan, if any.” This sentence might confuse the reader, a decrease in T₁ (increase in relaxation efficiency) indicates an increase in nanosecond motions.

Thanks for pointing out the ambiguous points, we have now corrected it to better distinguish the ^{13}C -T₁ time constant, ^{13}C -T₁ relaxation, and the characteristic nanosecond motion: “When all molecules were considered, the ^{13}C -T₁ relaxation time decreased in the order of cellulose, lignin, xylan, and mannan, if any. The short ^{13}C -T₁ time constants of lignin and hemicellulose revealed the efficient ^{13}C -T₁ relaxation in these non-cellulosic polymers, and furthermore, their enhanced motion on the nanosecond timescale.”

REVIEWERS' COMMENTS

Reviewer #1 (Remarks to the Author):

The authors responses and modification were acceptable. This is a really good piece of work.

Reviewer #2 (Remarks to the Author):

The authors have done a thorough revision with sufficient additional experimental data to address the comments I raised in my review.